# DCFold: Efficient Protein Structure Generation with Single Forward Pass

**Zhe Zhang**[1,2]    **Yuanning Feng**[1,3]    **Yuxuan Song**[1,4]    **Keyue Qiu**[1]    **Hao Zhou**[1] *    **Wei-Ying Ma**[1]

[1] Institute for AI Industry Research (AIR), Tsinghua University
[2] Department of Computer Science and Technology, Tsinghua University
[3] School of Computer Science and Technology, Huazhong University of Science and Technology
[4] ByteDance Seed

## Abstract

AlphaFold3 introduces a diffusion-based architecture that elevates protein structure prediction to all-atom resolution with improved accuracy. This state-of-the-art performance has established AlphaFold3 as a foundation model for diverse generation and design tasks. However, its iterative design substantially increases inference time, limiting practical deployment in downstream settings such as virtual screening and protein design. We propose DCFold, a single-step generative model that attains AlphaFold3-level accuracy. Our Dual Consistency training framework, which incorporates a novel Temporal Geodesic Matching (TGM) scheduler, enables DCFold to achieve a **15×** acceleration in inference while maintaining predictive fidelity. We validate its effectiveness across both structure prediction and binder design benchmarks.

## 1 Introduction

Proteins realize their biological functions through intricate three-dimensional conformations, and predicting such structures has long been a central challenge in computational biology. AlphaFold2 marked a breakthrough by combining multiple sequence alignments with geometric constraints in an end-to-end framework, achieving near-experimental accuracy (Jumper et al., 2021). Building on this foundation, AlphaFold3 reformulates the architecture into an all-atom framework and introduces a diffusion-based structure module, thereby enabling the generative modeling of not only proteins but also a wide spectrum of biomolecular complexes (Abramson et al., 2024). Consequently, this series of models are widely adopted as foundation models for downstream applications such as virtual screening and protein design (Alhumaid & Tawfik, 2024;

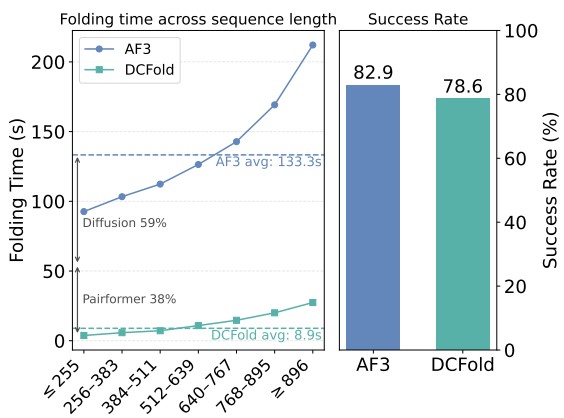

Figure 1: The acceleration ratio and generative quality of DCFold on Posebusters V2.

Baselious et al., 2024; Jendrusch et al., 2025; Frank et al., 2024; Bennett et al., 2023). However, AlphaFold3's architecture, which relies on iterative Pairformer recycling and multi-step diffusion (Ho et al., 2020), requires substantially greater computational overhead than AlphaFold2, restricting its accessibility in downstream workflows.

More specifically, we observe that on long sequences, the execution time of AlphaFold3 is measured in minutes, which severely limits its usability in downstream tasks that demand high throughput. For instance, small-scale laboratory screening often requires predictions for thousands of candidates (Li et al., 2023), and when extended to large public databases, this number grows to an infeasible

---

*Correspondence to Hao Zhou (zhouhao@air.tsinghua.edu.cn).

scale; protein design tasks typically involve comparable computational demand. Previous work such as BindCraft has attempted to mitigate this by manually reducing the number of recycling iterations on simpler structures, thus trading accuracy for efficiency (Pacesa et al., 2024). However, such compromises inevitably degrade predictive performance. Moreover, in hallucination-based approaches, the multistep iterative refinement process hinders feasible gradient backpropagation, ultimately preventing the broader community from adopting AlphaFold3 as a foundation model for diverse applications.

To accelerate the diffusion process, recent advances in generative modeling have explored the use of high-order solvers and consistency models. While high-order solvers improve efficiency, they rarely reduce the number of sampling steps below 10 (Lu et al., 2022; Zhao et al., 2023). Consistency models, on the other hand, have achieved remarkable success in image generation and benefited from refined training schedules (Song et al., 2023; Song & Dhariwal, 2023; Lu & Song, 2024). However, directly applying them to AlphaFold3 faces two major challenges: (i) standard schedules assume fixed-dimensional data and pair steps by a constant Euclidean distance, which fails to accommodate variable protein sequence lengths and leads to unstable training dynamics (details in Section 4.4); and (ii) AlphaFold3's architecture also relies on iterative Pairformer recycles, introducing an additional bottleneck that conventional diffusion consistency methods cannot address.

To address these challenges, we propose DCFold, a single-step folding model trained under Dual Consistency framework that attains AlphaFold3-level accuracy. We mitigate the inference bottleneck by jointly enforcing Pairformer Consistency and Diffusion Consistency, thereby eliminating both sources of iterative overhead. Crucially, we address the fundamental challenge of diffusion acceleration through rigorous theoretical derivations, and subsequently introduce a novel Temporal Geodesic Matching (TGM) scheduler, which adaptively pairs timesteps in the intrinsic geometric space of proteins. Together, these innovations preserve the predictive accuracy of AlphaFold3 while drastically reducing inference costs, enabling one-step predictions that are both efficient and reliable. We extensively validate the effectiveness of DCFold on structure prediction benchmarks, which provide a rigorous and fair evaluation protocol. Beyond this standard setting, we further assess DCFold in the more practical binder design tasks, where both inference speed and structural accuracy are critical to this setting.

In short, we summarize our contributions as follows:

- We propose DCFold, an inference-efficient structure prediction model that achieves performance and flexibility comparable to state-of-the-art applications. By leveraging the Dual Consistency framework, DCFold eliminates the iterative overhead inherent in AlphaFold3's architecture.

- We identify the key limitations of conventional consistency model (CM) methods when applied to variable-length protein sequences, and introduce Temporal Geodesic Matching (TGM) for a novel consistency schedule that both stabilizes training and yields improved performance.

- We evaluate the performance of DCFold across a diverse set of benchmarks and settings. On both Posebusters V2 and Recent PDB, it reaches AlphaFold3-level accuracy while achieving a notable $15\times$ speedup. Implemented in the binder design pipeline, DCFold demonstrates strong foundational capabilities while employing a lightweight architecture that ensures feasible gradient propagation. This design significantly improves the success rate of in silico screening by enabling faster and more reliable candidate evaluation.

## 2 PRELIMINARY

Diffusion models have emerged as a powerful class of generative models, achieving state-of-the-art performance across image, audio, and molecular generation tasks (Ho et al., 2020; Rombach et al., 2022; Trippe et al., 2022). A key limitation of standard diffusion samplers is their reliance on dozens to hundreds of function evaluations, which renders inference prohibitively expensive in high-dimensional settings such as protein folding. To address this bottleneck, recent work has focused on diffusion acceleration, aiming to distill or redesign the sampling process into far fewer steps. Among these approaches, *Consistency Models* (CMs) (Song et al., 2023) provide a principled framework built upon the probability flow ODE (PF-ODE), which establishes a bijective mapping between the clean data distribution and the noise distribution. CMs introduce a consistency function $f_\theta(x_t, t)$ that directly maps a noisy sample $x_t$ at time $t$ back to the clean signal $x_0$, subject to the boundary condition $f_\theta(x_0, 0) = x_0$. Training then proceeds by discretizing the PF-ODE into a

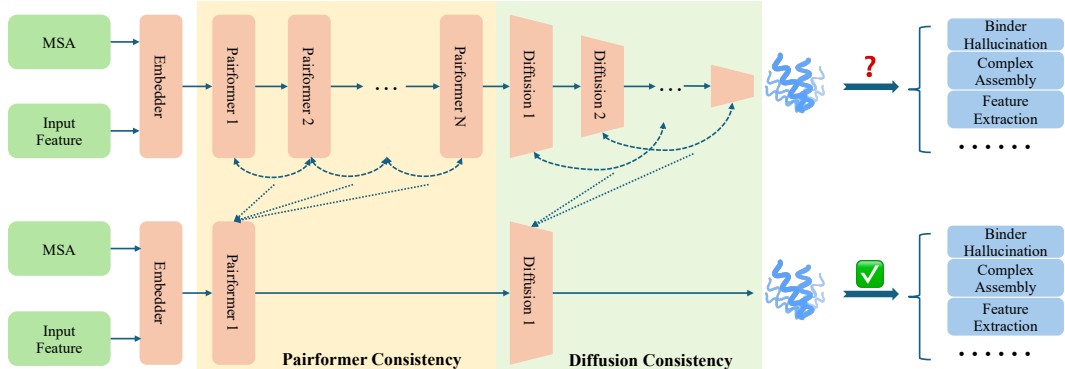

Figure 2: Overview of Dual Consistency framework (top: AlphaFold3; bottom: DCFold).

curriculum of time intervals $t_i$, and minimizing a loss that enforces functional consistency across adjacent timesteps,

$$\mathcal{L}_{\text{CM}} = \mathbb{E}\left[w(t_i)d\left(f_\theta(x_{t_{i+1}}, t_{i+1}), f_{\theta^-}(\tilde{x}_{t_i}, t_i)\right)\right], \tag{1}$$

where $w : \mathbb{R}_{\geq 0} \rightarrow \mathbb{R}^+$ denotes a positive weighting function, $d(\cdot, \cdot)$ is a metric function, $\theta^-$ is an EMA copy of the network, and $\tilde{x}_{t_i}$ is obtained by one-step PF-ODE integration. This objective ensures that the model predictions are invariant to the choice of sampling timestep, thereby collapsing multi-step trajectory into a single-step or few-step generator. Building on this foundation, subsequent refinements such as iCT (Song & Dhariwal, 2023), sCM (Lu & Song, 2024), and ECM (Geng et al., 2024), have optimized the weighting functions, discretization schedules, and training methodologies, resulting in substantial improvements in both efficiency and sample quality.

## 3 METHOD

### 3.1 OVERVIEW

We introduce DCFold, a high-accuracy single-step predictor. In Section 3.2, we describe the components of the Dual Consistency framework, which enforces consistency across the two major bottlenecks of AlphaFold3. In Section 3.3, we zoom in on the diffusion acceleration challenge and identify the key issue with prior consistency-based methods when training on variable-length sequences within diffusions. To tackle this challenge for complex structure prediction, we propose Temporal Geodesic Matching (TGM), which stabilizes training on the protein sequence modality.

### 3.2 DUAL CONSISTENCY

We identify the major factors impeding AlphaFold3's inference efficiency as the iterative diffusion process and Pairformer recycling, as illustrated in Figure 1. To address the first challenge, we investigate the behavior of AlphaFold3 under few-step sampling and find that its failure primarily arises from the sampling procedure

| Stage | Module | $\mathcal{L}_{\text{confidence}}$ | $\mathcal{L}_{\text{diffusion}}$ | $\mathcal{L}_{\text{pairformer}}$ |
|-------|--------|------------------|-----------|------------|
| (i)   | Diffusion  | $10^{-4}$ | 1 | $\times$ |
| (ii)  | Pairformer | $10^{-4}$ | $\times$ | 1 |

Table 1: Training stages and the weights of each term.

itself. The default strategy of injecting extra stochastic noise and enlarging the ODE step size turns out to be detrimental in this regime: the enlarged step size significantly amplifies the bias in ODE predictions. To stabilize performance, we modify the sampler by disabling noise injection (setting the noise factor $\gamma_0 = 0$), fixing the rescaling factor $\lambda = 1$, and normalizing the step size with $\eta = 1$, thereby enabling stable one-step sampling.

The first challenge concerns computational efficiency. After enabling one-step sampling, the Pairformer becomes the critical bottleneck. To tackle this, we introduce **Dual Consistency**, which applies consistency learning to both the diffusion module and the Pairformer.

**Diffusion Consistency** Although we already have a functional one-step sampler, we aim to maximize its utility. Specifically, we apply consistency distillation to the diffusion module, aligning its

---

**Algorithm 1** Temporal Geodesic Matching (TGM)

---

**Require:** Dataset $\mathcal{D}$, pretrained diffusion model $\theta$, noise distribution $p(t)$, weighting function $w(t)$,
    training progress $u = \frac{\text{steps}}{\text{max\_steps}} \in [0,1]$

 1: **while** $\theta$ not converged **do**
 2:     Sample $x_0 \sim \mathcal{D}, \epsilon \sim \mathcal{N}(0, I), t \sim p(t)$
 3:     $r' \leftarrow \max\left(r(t, u), 0\right)$
 4:     $x_t \leftarrow x_0 + t\epsilon; \quad x_{r'} \leftarrow x_0 + r'\epsilon$
 5:     $\mathcal{L} \leftarrow w(t) \|f_\theta(x_t, t) - f_{\text{sg}(\theta)}(x_{r'}, r')\|_2^2$            ▷ using the same random seed
 6:     $\theta \leftarrow \theta - \eta\nabla_\theta\mathcal{L}$
 7: **end while**

---

single-step performance with that of the multi-step counterpart, which also provides a natural warm-up for the subsequent Pairformer consistency stage. The training objective minimizes the MSE between the outputs of the diffusion module at timestep $t$ and a reference timestep $r$. Formally, the diffusion consistency loss is

$$\mathcal{L}_{\text{diffusion}} = \mathbb{E}_{x,t,r,\epsilon}\left[w(t)\text{MSE}\left(f_\theta(x_t, t) - f_{\text{sg}(\theta)}(x_r, r)\right)\right], \tag{2}$$

where $f_\theta$ denotes diffusion module parameterized by $\theta$, and $\text{sg}(\theta)$ denotes "stop-gradient" operator. We find $w(t)$ to have negligible effect in experiments and therefore set $w(t) = 1$.

**Pairformer Consistency**  For the most critical bottleneck in AlphaFold3, Pairformer, we observe that the architecture updates internal protein representations iteratively across multiple cycles. While increasing the number of cycles generally improves prediction accuracy, it also scales inference time linearly. Importantly, because each Pairformer cycle depends on the output of the previous one, a single forward pass through the network inherently provides representations corresponding to different cycle depths. This allows us to assess the model's progressive refinement of structural accuracy without the need for explicit time sampling as required in diffusion-based denoising processes.

To exploit this property, we introduce a **cycle consistency loss**. Suppose pairformer runs for $N$ cycles (with $N = 4$ in our experiments). After the $n$-th cycle, the model produces a pair representation $z_n$ and a single representation $s_n$. We directly adopt the *total transmission error* as the loss function:

$$\mathcal{L}_{\text{pairformer}} = \sum_{i=1}^{N-1}\left(\text{MSE}\left(z_i, z_{i+1}\right) + \text{MSE}\left(s_i, s_{i+1}\right)\right). \tag{3}$$

Notably, we adopt the weighting strategy from AlphaFold's supervised MSE loss. In particular, positions corresponding to nucleic acids and small molecules are assigned higher weights than amino acids. This ensures that structurally sensitive residues contribute proportionally to the loss. Let the column vector $\boldsymbol{\alpha}$ denote the per-token weighting coefficient used in AlphaFold3. For the single representations in both Diffusion Consistency and Pairformer Consistency, we directly apply $\boldsymbol{\alpha}$ as the weight. In contrast, for the pair representations in Pairformer, we adopt a multiplicative composition, using $\sqrt{\boldsymbol{\alpha}}\sqrt{\boldsymbol{\alpha}}^\top$ as the weighting matrix, where the square root is applied element-wise.

We further find that incorporating the confidence loss $\mathcal{L}_{\text{confidence}}$ from AlphaFold3's confidence head improves training stability, where $\mathcal{L}_{\text{confidence}}$ is defined as:

$$\mathcal{L}_{\text{confidence}} = \mathcal{L}_{\text{plddt}} + \mathcal{L}_{\text{pde}} + \mathcal{L}_{\text{resolved}} + \alpha_{\text{pae}} \cdot \mathcal{L}_{\text{pae}},$$

where $\alpha_{\text{pae}} = 1$, and the definitions of all loss terms follow AlphaFold3. Consequently, our training procedure can be summarized in two stages: (i) train a one-step sampler, where only the diffusion module is updated, with the training objective given by $\mathcal{L}_{\text{confidence}}$ and $\mathcal{L}_{\text{diffusion}}$; (ii) apply pairformer consistency, where only a 16-block Pairformer is updated, with the training objective given by $\mathcal{L}_{\text{confidence}}$ and $\mathcal{L}_{\text{pairformer}}$. We summarize the weights of them in Table 1.

### 3.3 Temporal Geodesic Matching

While consistency-based methods have shown promise, directly applying them to complex architectures like AlphaFold often results in weight collapse, high training cost, or reliance on task-specific mappings. The core issue lies in scheduling for variable-size outputs such as protein structures. Conventional schedulers pair timesteps $(t, r)$ at fixed Euclidean intervals, producing an ill-posed

curriculum: on long sequences, even small $\Delta t$ triggers drastic distribution shifts that demand unrealistic predictive leaps, whereas on short sequences the same interval provides only weak signals. This mismatch overlooks the non-uniform accumulation of information with data dimensionality, leading to instability and collapse.

To address these limitations, we introduce Temporal Geodesic Matching (TGM), a general and scalable distillation framework. TGM explicitly selects training pairs $(t, r)$ such that their geodesic distance on the temporal information manifold is preserved, thereby offering a principled mechanism to stabilize training and extend consistency learning to large-scale protein modeling tasks. Unlike Euclidean-based heuristics, TGM aligns the distillation dynamics with the intrinsic statistical geometry of the diffusion trajectory. By doing so, it ensures stability and fidelity even in high-dimensional structured output spaces such as protein backbones.

We begin by formalizing the diffusion trajectory as a geometric object. Let $p_t(x)_{t \in [0,T]}$ denote the family of intermediate distributions induced by the forward diffusion process. We interpret it as a coordinate charting a one-dimensional **temporal information manifold** $\mathcal{M}_t$, where each point corresponds to a distribution $p_t(x)$.

**Definition 1** *We define the temporal metric via the Fisher information with respect to the diffusion time $t$, which we refer to as the **temporal Fisher information**, and use it as the Riemannian metric tensor of $\mathcal{M}_t$:*

$$g(t) := \mathcal{I}(t) = \mathbb{E}_{p_t(x)} \left[ \left( \frac{\partial}{\partial t} \log p_t(x) \right)^2 \right]. \tag{4}$$

**Definition 2** *On the manifold where the temporal Fisher information serves as the Riemannian metric tensor, the **geodesic distance** between two time points $t$ and $r$ is defined as the corresponding geodesic length:*

$$d_g(t, r) = \int_r^t \sqrt{\mathcal{I}(\tau)} d\tau. \tag{5}$$

Our central thesis is that a stable and efficient distillation process must be grounded in the Kullback-Leibler (KL) divergence, as this is the canonical metric underlying the variational objective of diffusion models. We motivate the introduction of the Fisher information through the following theorem:

**Proposition 1** *(Local Metric-KL Equivalence) For a small step $\Delta t = t - r \geq 0$, the geodesic distance between neighboring distributions is given by:*

$$d_g(t, r) = \sqrt{2} D_{\mathrm{KL}} \left( p_r(x) \| p_t(x) \right)^{\frac{1}{2}} + \mathcal{O} \left( (\Delta t)^3 \right). \tag{6}$$

The proof of Proposition 1 is provided in the Appendix A.1. The metric $d_g$ provides a principled measure of distributional discrepancy along the temporal axis. Building on this, TGM stabilizes training by enforcing a consistent alignment rule: for a given training progress $u = \frac{\text{steps}}{\text{max\_steps}} \in [0, 1]$, each timestep $t$ is paired with a reference point $r$ at a fixed temporal distance, i.e., $d_g(t, r) = C(u)$, where $C(u)$ is a monotonically decreasing function. In our experiments, we specify $C(0) = C_0$ as a hyperparameter, $C(1) = 0, C(u) = C_0(1 - u)^\beta, \beta > 0$, and approximate $r(t, u) = t - \frac{C_0}{\sqrt{\mathcal{I}(t)}} (1 - u)^\beta$ via one-step Euler method. While it is also feasible to employ higher-order numerical solvers, we did not observe significant performance gains from doing so. Furthermore, we provide the analytical form of $\mathcal{I}(t)$:

**Proposition 2** *For any diffusion model that satisfies the classical setting of $p_t(x|x_0) = \mathcal{N}(x; \mu = \alpha(t)x_0, \sigma^2(t)I)$:*

$$\mathcal{I}(t) = \mathbb{E}_{x_0 \sim p_{\text{data}}} \left[ \frac{\dot{\sigma}(t)}{\sigma(t)} \cdot 2D + \frac{\dot{\alpha}(t)}{\sigma(t)} \|x_0\|^2 \right], \tag{7}$$

*where $D$ denotes the dimensionality of the vector.*

This analytical form underscores the universality of TGM. In most generative tasks, data can naturally be represented as fixed-length vectors. Furthermore, when normalized (as in image generation) or invariant to random rotations (as in protein folding), the $\|x_0\|^2$ term admits a simplification to $\text{Var}(x_0)$ under the assumption $\mathbb{E}[x_0] = 0$. In our experiments, due to AlphaFold's adoption of the EDM framework(Karras et al., 2022), we present here the specific form of $\mathcal{I}(t)$ that is used:

$$\mathcal{I}(t) = \frac{2D \cdot p \left( s_{\max}^{1/p} - s_{\min}^{1/p} \right)}{s_{\max}^{1/p} + (1 - t) \left( s_{\min}^{1/p} - s_{\max}^{1/p} \right)}, \tag{8}$$

Table 2: Posebusters V2 RMSD benchmark results. We report the percentage of predictions with RMSD below different thresholds.

| Method | Best (%) | | | | Worst (%) | | | |
|---|---|---|---|---|---|---|---|---|
| | $< 1$ | $< 2$ | $< 3$ | $< 5$ | $< 1$ | $< 2$ | $< 3$ | $< 5$ |
| AlphaFold3 | **67.14** | **82.86** | **87.14** | 93.81 | 45.71 | 70.00 | 79.05 | 87.62 |
| AF3 ODE | 51.43 | 74.77 | 83.81 | 92.38 | 37.62 | 66.19 | 75.71 | 87.62 |
| DCFold (Ours) | 58.10 | 78.57 | 86.67 | **94.29** | **46.67** | **71.43** | **80.00** | **90.48** |

Table 3: TM-score and Success Rate (SR) on different protein categories in the Homology Recent PDB dataset. Values in parentheses denote the absolute improvement relative to AF3 ODE.

| Method | PL-complex | | Monomer | | PP-complex | |
|---|---|---|---|---|---|---|
| | TM-score | SR (%) | TM-score | SR (%) | TM-score | SR (%) |
| AF3 ODE | 0.815 | 92.3 | 0.830 | 92.9 | 0.763 | 87.0 |
| AlphaFold3 | 0.810 (–0.6) | 93.9 (+1.6pp) | 0.839 (+1.0) | 94.5 (+1.6pp) | 0.788 (+3.2) | 91.1 (+4.0pp) |
| DCFold (Ours) | **0.824 (+1.2)** | **94.9 (+2.6pp)** | **0.850 (+2.3)** | **95.7 (+2.9pp)** | **0.800 (+4.8)** | **92.2 (+5.2pp)** |

where the definition of $s_{\min}$ and $s_{\max}$ follow EDM, which are used in AlphaFold3's diffusion process to control the noise strength. Here we incorporate the data dimensionality $D$ into the training schedule to balance the differences in learning difficulty across amino acid sequences of varying lengths. Importantly, as the dimensionality increases, the KL divergence between distributions accumulates linearly, causing classical consistency training to exaggerate information disparities for long sequences. And we provide in Algorithm 1 the procedure for applying TGM to the diffusion module.

## 3.4 Downstream Task

After ensuring the consistency of AlphaFold3, we find that our method now holds substantial potential for downstream applications. As a representative example, we validate the effectiveness of DCFold in the task of binder design. This task typically requires models to perform large-scale sampling, followed by stringent multi-stage filtering to eliminate implausible sequences, leaving only a small subset of viable candidates. Moreover, in binder hallucination–based design frameworks, the network must be fully differentiable and amenable to gradient-based optimization (Pacesa et al., 2024). These properties make DCFold particularly well-suited for this setting, allowing it to fully demonstrate its performance advantages. The experimental details are presented in Section 4.3.

## 4 Experiment

We design our experiments to evaluate both the accuracy and practical utility of DCFold. In Section 4.1, we evaluate the structural prediction capability of DCFold, showing that DCFold matches or surpasses AlphaFold3 while reducing cost. In Section 4.3, we assess binder hallucination, demonstrating that the reshaped output distribution improves downstream design success. Section 4.4 isolates the effect of TGM and shows its advantage over prior consistency schedules. Together, these results highlight the efficiency, stability, and applicability of DCFold across protein modeling tasks.

## 4.1 Structure Prediction

In this section, we demonstrate that DCFold retains strong capability for one-step prediction.

**Baselines** We compare these AlphaFold3 variants: (i) **AlphaFold3** (Abramson et al., 2024) – The original configuration of AlphaFold3 employs the full set of recycling cycles and diffusion steps, serving as a strong baseline as well as the reference target that DCFold aims to approximate. (ii) **AF3 ODE** – AlphaFold3 configured with a single sampling step and a single recycling cycle, serving as a reference baseline without retraining. (iii) **AF3 TGM** – a partially distilled AlphaFold3 variant, which builds upon AF3 ODE by applying only our TGM diffusion consistency distillation without pairformer distillation. This isolates the contribution of TGM to performance under one-step sampling. (iv) **DCFold** – our fully distilled model after applying dual consistency training,

which uses only 1 recycle and 1 diffusion denoising step. Both the baseline and the initialization of DCFold are derived from Protenix, an open-source reimplementation of AlphaFold3. (v) **Protenix-Mini** – We also include a lightweight variant of Protenix, which reduces the parameter count from 368M to 135M and uses 2-step ODE sampling to lower computational cost.

**Data**  For training, we use PDB entries released after September 30, 2021, organized following the Protenix scheme with identical filtering protocols. Evaluation is performed on two benchmarks: (i) **PoseBusters V2** (Buttenschoen et al., 2024), a curated benchmark of recent high-quality protein–ligand crystal complexes with drug-like molecules, restricted to post-2021 releases; and (ii) the **Low Homology Recent PDB dataset** (Jumper et al., 2021; Team et al., 2025), containing numerous protein and nucleic acid interfaces. Introduced in AlphaFold3, we employ the Protenix open-source implementation. All entries predating the training cutoff are excluded from evaluation.

**Metrics**  On Posebusters V2, we evaluate predictions using the RMSD between predicted and experimental ligand coordinates. For each complex, we report the proportions of generated poses whose best and worst RMSDs (with respect to the ground-truth structure) fall below the thresholds of 1, 2, 3, and 5 Å. Ground truth is not used for any filtering, so this does not introduce data leakage. These metrics quantify how Dual Consistency reshapes AlphaFold3's output distribution. On RecentPDB, we measure backbone accuracy using the TM-score (Biasini et al., 2013), where values above 0.5 indicate correct folds; the success rate is defined as the proportion of structures with RMSD $< 2$ Å; and local accuracy is assessed using lDDT (Mariani et al., 2013), which ranges from 0–100 and reflects residue-level geometric precision.

Overall, DCFold achieves accuracy comparable to AlphaFold3 while using only a single recycle and diffusion step, demonstrating both efficiency and robustness. The results in Table 2, Table 3 and Figure 3 highlight these key observations:

**AlphaFold3 admits single-step generation.** With a proper choice of ODE parameters, the AF3 ODE solver is capable of generating approximately correct protein structures.

**DCFold enhances generative performance.** Training with Dual Consistency substantially improves the performance of the AF3 ODE model: across several RMSD thresholds, DCFold approaches or even matches AlphaFold3, demonstrating that the distilled model effectively recovers accuracy despite relying on only a single recycle and diffusion step.

**DCFold reshapes the distribution of generated structures.** Dual Consistency reshapes the output distribution of AlphaFold3 by effectively tightening it. This effect is reflected in the improved *worst*-case RMSD, indicating more stable and reliable predictions, while the *best*-case RMSD remains largely unchanged. Such a redistribution reduces extreme errors and enhances the consistency of single-step predictions, which is particularly valuable for accelerating downstream scientific workflows where both efficiency and reliability are critical.

The improvement is especially evident in Success Rate, where DCFold achieves substantially larger gains than in average TM-score. This observation further supports our claim that DCFold reshapes the distribution of generated structures. In particular, DCFold demonstrates a stronger ability than AlphaFold3 to avoid generating implausible biological complexes.

**Both components of Dual Consistency are beneficial.** In the lDDT experiments shown in Figure 3, DCFold delivers accuracy on par with AlphaFold3. We further conduct ablation studies disentangling the effects of Diffusion Consistency and Pairformer Consistency, and find that both components contribute complementary gains. Together, these results highlight that Dual Consistency is the key driver behind the observed improvements.

## 4.2  DIVERSITY AND CONFIDENCE

To more comprehensively characterize the performance of DCFold , we conducted an extended analysis of its structural diversity and predictive confidence on the Posebusters V2 benchmark.

**Metrics.**  For each test sequence, we sampled five structures and computed all pairwise TM-scores among these predictions. We report the dataset-level average of these pairwise values as the *Diversity* metric (lower is better). We further compute the mean pLDDT across all sampled structures as the *Confidence* metric (higher is better).

**DCFold maintains strong sample diversity and confidence.** As shown in Table 4, after Dual Consistency training, DCFold exhibits no substantial deviation from AlphaFold3 in either metric. Diversity shows a slight decrease, whereas confidence displays a slight increase. These trends sug-

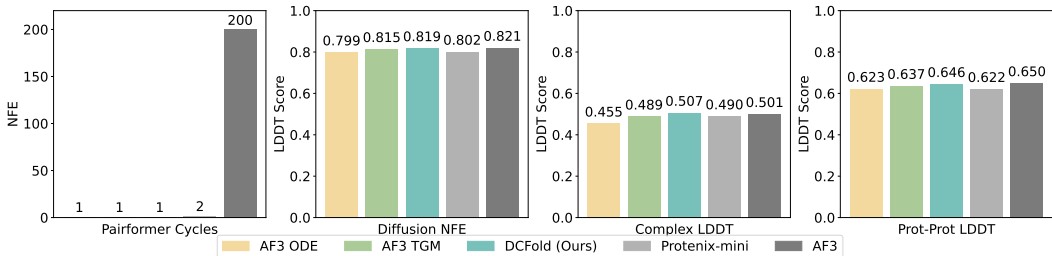

Figure 3: lDDT performance on the Recent PDB dataset.

Table 4: Diversity and confidence metrics on the Posebusters V2 benchmark.

| Method | Diversity ($\downarrow$) | Confidence ($\uparrow$) |
|---|---|---|
| AF3 (5 samples) | $0.9646 \pm 0.0410$ | $93.97 \pm 2.92$ |
| AF3 (15 samples) | $0.9642 \pm 0.0415$ | $93.95 \pm 2.93$ |
| AF3 (5 seeds $\times$ 1 sample) | $0.9697 \pm 0.0421$ | $93.90 \pm 3.01$ |
| DCFold (5 samples) | $0.9701 \pm 0.0565$ | $94.14 \pm 2.97$ |
| DCFold (15 samples) | $0.9708 \pm 0.0567$ | $94.13 \pm 2.96$ |
| DCFold (5 seeds $\times$ 1 sample) | $0.9712 \pm 0.0570$ | $94.15 \pm 2.97$ |

gest that enforcing Dual Consistency mildly concentrates the structural distribution while preserving high prediction quality.

To assess the robustness of these observations under increased sampling, we additionally evaluated: (i) 15 samples drawn under a fixed seed, and (ii) 5 random seeds with 1 sample each. Under both settings, neither AlphaFold3 nor DCFold exhibited meaningful improvements in diversity. This behavior aligns with the well-known strong conditionality of AlphaFold-series models, which tends to limit diversity gains from additional sampling alone.

Importantly, DCFold remains compatible with a broad set of diversity-enhancing strategies proposed in prior work, including sampling MSAs, clustering or masking MSA columns, and tuning dropout rates (Wayment-Steele et al., 2024; Wallner, 2023; Kalakoti & Wallner, 2025). Our acceleration approach is orthogonal to these methods, and all such techniques can be directly applied to DCFold with expected diversity improvements comparable to those previously reported for AlphaFold3.

### 4.3 BINDER HALLUCINATION

After maintaining the consistency between Pairformer and Diffusion, DCFold achieves efficient inference and stable gradient backpropagation with modest computational cost. We focus on the binder hallucination task, which serves as a representative benchmark due to its stringent requirements: it demands a fully differentiable folding model, while the filtering stage eliminates a large fraction of implausible candidates. As a result, success in this setting critically depends on achieving efficient inference. Following the same hallucination strategy and filtering pipeline as BindCraft (details provided in Appendix B.2) (Pacesa et al., 2024), we leverage confidence scores and additional loss terms from DCFold as feedback signals for sequence evaluation. To ensure a fair comparison, folding constraints are consistently computed using the outputs of AlphaFold2, thereby avoiding potential numerical discrepancies in confidence calibration between DCFold and AlphaFold2.

**Data** We adopt the six representative entries from Cao et al. (2022) as the design targets, namely IL2-R$\alpha$, TrkA, H3, VirB8, ALK, and LTK. They span multiple functional categories, including receptors, enzymes, transcription factors, and bacterial proteins. They have been widely adopted in prior studies as common benchmarks for design and docking tasks. For each case, we restrict binder length to 55–65 residues and perform a continuous 48-hour hallucination run.

**Metrics** We compute the Success Rate using the same two filters as BindCraft. The model-based constraint is derived from AlphaFold2's confidence score, whereas the physics-based constraint relies on physical metrics obtained from Rosetta. Additional details are provided in Appendix B.2.

Table 5: In silico success rates across six targets for binder design (values shown as physics-based constraints / model-based constraints).

|  | IL-2R$\alpha$ | TrkA | H3 | VirB8 | ALK | LTK | Average |
|---|---|---|---|---|---|---|---|
| BindCraft | **.38**/.84 | .29/**.88** | .16/.52 | .15/.72 | **.14**/.48 | .43/.70 | .26/.69 |
| DCFold (Ours) | .37/.79 | **.31**/.84 | **.23**/**.71** | **.21**/**.85** | .12/**.54** | **.47**/**.93** | **.29**/**.78** |

DCFold achieves higher in silico success rates than the AF2-based BindCraft baseline across the majority of targets. With the incorporation of DCFold, AlphaFold3 can readily support binder hallucination strategies that were previously only feasible within the AlphaFold2 framework. Notably, DCFold achieves much higher success rates on several targets (e.g., H3, VirB8, and LTK), indicating that our reshaping of AlphaFold3's output distribution translates into tangible improvements in downstream design tasks. These findings highlight that DCFold bridges the methodological gap between AlphaFold2- and AlphaFold3-based pipelines, and unlocks additional performance gains. We have added more details about the experimental results in Appendix C.2. Figure 7 visualizes representative binder–target complexes, illustrating the interactions between the generated binders and their targets.

## 4.4 EMPIRICAL VALIDATION OF TGM

We conduct experiments on feasible generic consistency-model baselines, including CD (Song et al., 2023), sCM (Lu & Song, 2024), ECM (Geng et al., 2024), and TGM. Results on Posebusters V2 are summarized in Table 6.[1] We observe that among all runnable baselines, a naive implementation of CD leads to training collapse and severely degrades performance. Only ECM and TGM are able to enhance the performance of the diffusion module, with TGM yielding the largest performance gains.

Table 6: Success Rates of Different Consistency Models on Posebusters V2.

| Method | Time (s/step) | Success rate (%) |
|---|---|---|
| CD | 18.5 | 25.6↓ |
| sCM | 38.1 | - |
| ECM | **11.6** | 75.7↑ |
| TGM | **11.6** | **77.5**↑ |

Therefore, in the following experiments, we take ECM as the representative of prior general consistency models and investigate how TGM exhibits distinct behavior on protein folding tasks. Detailed hyperparameter settings for each method are provided in Appendix B.3.

We conduct an in-depth analysis of the sources of improvement introduced by TGM and present the gradient norm and loss curve throughout training in Figure 5. We observe that the training dynamics of ECM exhibits poor smoothness, characterized by distinct staircase-like patterns, and is accompanied by a large gradient variance. This corroborates our hypothesis in Section 3.3 that classical consistency algorithms degrade under variable-length sequences. In contrast, TGM consistently maintains balanced gradients, indicating that the learning difficulty of the network remains at a fixed distance from its current capacity, effectively counteracting the adverse effects introduced by variable-length sequences.

In addition, we further assess whether the Euler method employed in TGM introduces excessive numerical error in Figure 4. We observe that the error is relatively large during the early stages of training but decreases as training progresses, leading to more accurate estimates in later

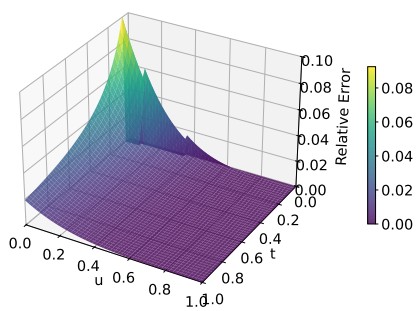

Figure 4: The relative error of the Euler solver for $r(t, u)$.

stages. Moreover, the error remains consistently low throughout the entire training process, indicating that our approximation is sufficiently reliable. This also explains why employing higher-order algorithms does not yield substantially greater benefits.

---

[1]Due to the substantial computational overhead of sCM, processing long sequences often results in out-of-memory (OOM) errors, preventing it from participating in a fair comparison.

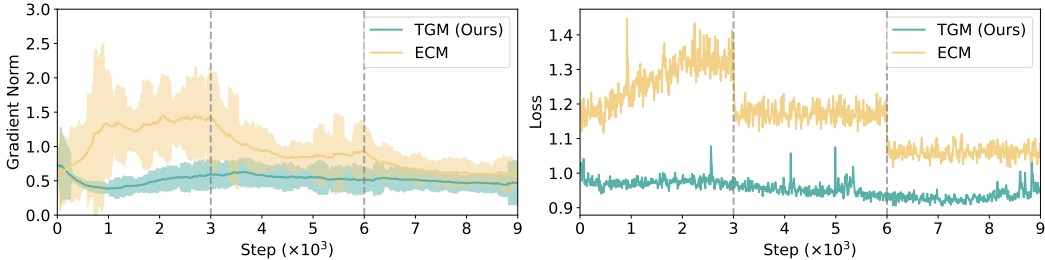

Figure 5: Gradient norm and loss curve during training for ECM and TGM.

## 5 RELATED WORK

**Protein Structure Prediction**    Protein structure prediction has rapidly advanced with deep learning. Classical methods such as Rosetta (Rohl et al., 2004) and co-evolutionary analysis (Marks et al., 2011; Ovchinnikov et al., 2017) provided key insights but were limited in accuracy and scalability. The advent of deep neural networks enabled models like RaptorX (Xu, 2019) and trRosetta (Yang et al., 2020) to exploit large multiple sequence alignments (MSAs), setting the stage for a decisive breakthrough. AlphaFold2 (Jumper et al., 2021) combined evolutionary information with a novel attention architecture, achieving near-experimental resolution.

Efforts to reduce reliance on MSAs led to models such as ESMFold (Lin et al., 2022), OmegaFold (Wu et al., 2022), and HelixFold-Single (Fang et al., 2022), which leverage protein language models for fast single-sequence prediction, albeit at lower accuracy. Extensions like AlphaFold-Multimer (Evans et al., 2021) generalized AF2 to protein–protein interactions, establishing it as a foundation model. Building on this, AlphaFold3 (Abramson et al., 2024) introduced a diffusion-based structure module and unified biomolecular representation, enabling prediction of protein–ligand, nucleic acid, and heterogeneous complexes. Despite setting new standards in accuracy and scope, AF3's computational overhead remains a key barrier, driving research into acceleration, distillation, and approximation (Cheng et al., 2022).

**Diffusion Acceleration**    Recent advances in diffusion acceleration fall into three categories: training-free solvers, training-based distillation, and flow-based reformulations. Training-free solvers leverage higher-order integration, predictor–corrector schemes, and adaptive noise schedules to achieve high-quality generation in a few dozen steps, though performance often degrades in the extreme few-step regime (Song et al., 2020; Lu et al., 2022; Zhao et al., 2023). Training-based distillation compresses long diffusion chains into compact generators: progressive distillation iteratively reduces step counts, adversarial variants integrate GAN-style objectives, and Consistency Models (CMs) enforce self-consistency across time to enable single- or few-step generation with strong fidelity (Salimans & Ho, 2022; Sauer et al., 2024; Song et al., 2023). In parallel, flow-based methods reformulate diffusion as velocity fields with straightened trajectories, allowing efficient integration with simple solvers (Liu et al., 2022; Lipman et al., 2022).

## 6 CONCLUSION

We present DCFold, a dual-consistency distillation framework that compresses AlphaFold3 into a high-fidelity single-step sampler. By jointly enforcing diffusion and Pairformer consistency and introducing the Temporal Geodesic Matching schedule, DCFold achieves stable training on variable-length protein sequences while reducing inference cost by up to **15×**. Experiments on structure prediction and binder design show that DCFold matches or surpasses AlphaFold3 in accuracy and substantially improves downstream usability, bridging AlphaFold2's efficiency with AlphaFold3's accuracy to enable scalable, differentiable protein design.

## ETHICS STATEMENT

This work focuses on methodological contributions to protein structure prediction and design. All experiments are conducted on publicly available datasets such as the Protein Data Bank (PDB)

and established benchmarks, without involving human subjects, sensitive personal data, or animal studies. The proposed methods are intended solely for advancing computational biology research. Therefore, we do not identify any specific ethical concerns associated with this work.

## REPRODUCIBILITY STATEMENT

We disclose all training details in Section 3.2 and Section 3.3, enabling full reproducibility of our experimental results. Moreover, we will release both the pretrained weights and the source code to ensure transparency and facilitate future research.

## ACKNOWLEDGMENTS

The authors would thank the anonymous reviewers for reviewing the draft. This work is supported by the Natural Science Foundation of China (Grant No. 62376133), Beijing Nova Program (Grant No. 20240484682) and Wuxi Research Institute of Applied Technologies, Tsinghua University under Grant 20242001120.

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

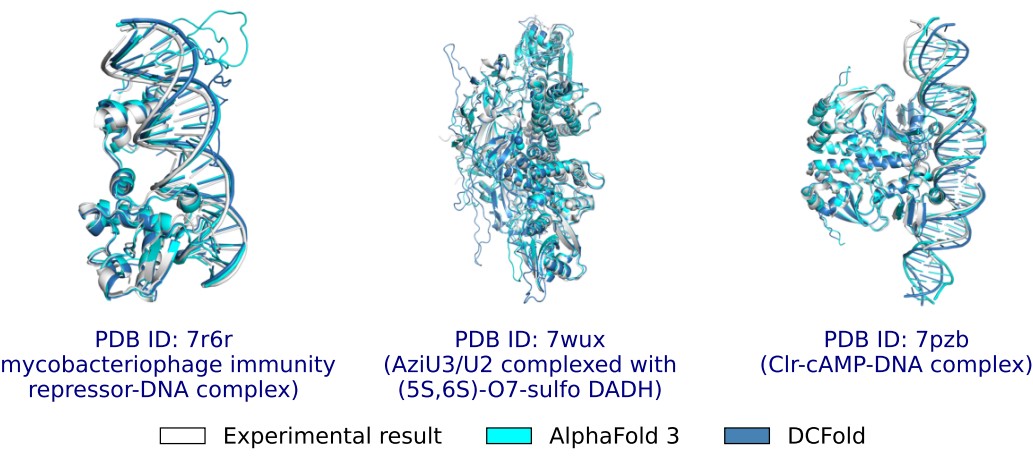

Figure 6: A structure prediction case study of DCFold, compared against AlphaFold3 and the experimental result.

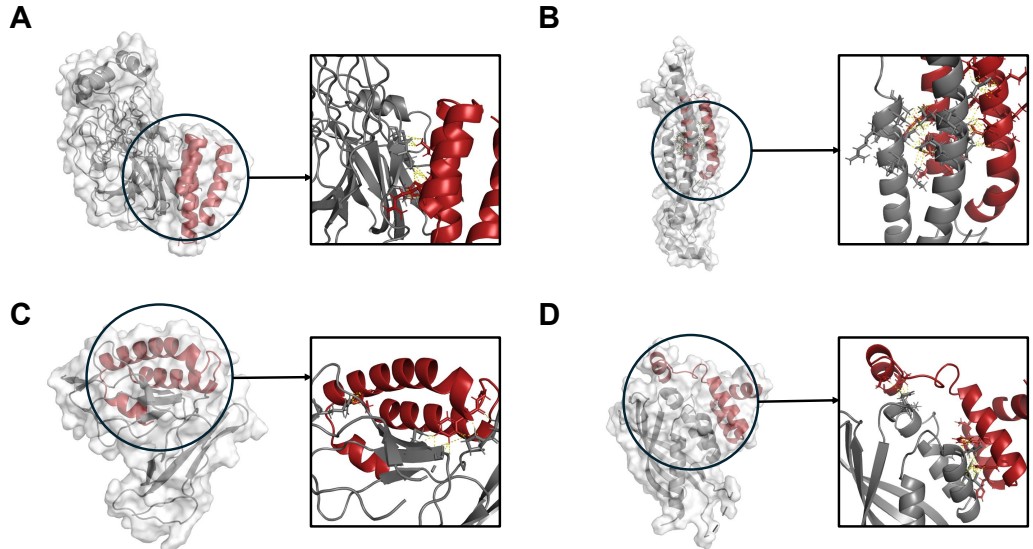

Figure 7: Examples from binder-design experiments, with targets: (A) ALK, (B) H3, (C) IL2R$\alpha$, and (D) VirB8.

## A  DERIVATION OF TGM

### A.1  PROOF OF LOCAL METRIC-KL EQUIVALENCE

We investigate the KL divergence between two distributions defined on the manifold $\mathcal{M}_t$:

$$D_{\mathrm{KL}}(p_r \| p_t) = \int p_r(x) \log \frac{p_r(x)}{p_t(x)} \, dx = \int p_r(x) \left[ \log p_r(x) - \log p_t(x) \right] \, dx \qquad (9)$$

We perform a Taylor expansion of $\log p_{t-\Delta t}(x)$ and substitute the result into the KL divergence.

$$\log p_{t-\Delta t}(x) = \log p_t(x) - \Delta t \frac{\partial}{\partial t} \log p_t(x) + \frac{1}{2} (\Delta t)^2 \frac{\partial^2}{\partial t^2} \log p_t(x) + \mathcal{O}\left((\Delta t)^3\right) \qquad (10)$$

Substituting it into the KL divergence yields:

$$D_{\mathrm{KL}}(p_r\|p_t) = \int p_r(x)\left[-\Delta t\frac{\partial}{\partial t}\log p_t(x) + \frac{1}{2}(\Delta t)^2\frac{\partial^2}{\partial t^2}\log p_t(x) + \mathcal{O}\left((\Delta t)^3\right)\right]dx$$

$$= \int\left[p_t(x) - \Delta t\frac{\partial}{\partial t}p_t(x) + \frac{1}{2}(\Delta t)^2\frac{\partial^2}{\partial t^2}p_t(x) + \mathcal{O}\left((\Delta t)^3\right)\right]\cdot$$

$$\left[-\Delta t\frac{\partial}{\partial t}\log p_t(x) + \frac{1}{2}(\Delta t)^2\frac{\partial^2}{\partial t^2}\log p_t(x) + \mathcal{O}\left((\Delta t)^3\right)\right]dx \tag{11}$$

The first-order term vanishes:

$$-\Delta t\int p_t(x)\frac{\partial}{\partial t}\log p_t(x)\,dx = 0, \tag{12}$$

while the second-order term takes the following form:

$$\frac{(\Delta t)^2}{2}\int p_t(x)\frac{\partial^2}{\partial t^2}\log p_t(x)\,dx + (\Delta t)^2\int\frac{\partial}{\partial t}p_t(x)\frac{\partial}{\partial t}\log p_t(x)\,dx \tag{13}$$

The term on the right-hand side is given by

$$(\Delta t)^2\int p_t(x)\left[\frac{\partial}{\partial t}\log_t(x)\right]^2 dx = (\Delta t)^2\mathcal{I}(t) \tag{14}$$

The simplification of the left-hand side relies on the property that the integral of the score function vanishes:

$$0 = \frac{\partial}{\partial t}\cdot 0 = \frac{\partial}{\partial t}\int p_t(x)\frac{\partial}{\partial t}\log p_t(x)\,dx = \int\frac{\partial}{\partial t}p_t(x)\frac{\partial}{\partial t}\log p_t(x)\,dx + \int p_t(x)\frac{\partial^2}{\partial t^2}\log p_t(x)\,dx \tag{15}$$

Thus, the term on the left-hand side can also be expressed in terms of $\mathcal{I}(t)$:

$$\int p_t(x)\frac{\partial^2}{\partial t^2}\log p_t(x)\,dx = -\int\frac{\partial}{\partial t}p_t(x)\frac{\partial}{\partial t}\log p_t(x)\,dx = -\mathcal{I}(t) \tag{16}$$

Thus, the second-order term implicitly encodes the temporal Fisher information $-\frac{(\Delta t)^2}{2}\mathcal{I}(t) + (\Delta t)^2\mathcal{I}(t) = \frac{(\Delta t)^2}{2}\mathcal{I}(t)$, that is $D_{\mathrm{KL}}\left(p_r(x)\|p_t(x)\right) = \frac{(\Delta t)^2}{2}\mathcal{I}(t) + \mathcal{O}\left((\Delta t)^3\right)$. With this, local metric-KL equivalence becomes evident.

## A.2 TEMPORAL FISHER INFORMATION IN EDM

We assume the forward process of diffusion is defined as $p_t(x|x_0) = \mathcal{N}(x;\mu = \alpha(t)x_0, \sigma^2(t)I), \mathcal{I}(t) = \mathbb{E}_{p_t(x)}\left[\left(\frac{\partial}{\partial t}\log p_t(x)\right)^2\right] = \mathbb{E}_{x_0\sim p_{\mathrm{data}}}\mathbb{E}_{p_t(x|x_0)}\left[\left(\frac{\partial}{\partial t}\log p_t(x)\right)^2\right]$

We employ a multivariate Gaussian distribution with dimensionality $D$: $p(x) = \frac{1}{(2\pi)^{D/2}|\Sigma|^{1/2}}\exp\left(-\frac{1}{2}(x-\mu)^\top\Sigma^{-1}(x-\mu)\right)$, $\Sigma = \sigma^2(t)I, |\Sigma| = |\sigma^2(t)I| = \sigma^{2D}(t), \Sigma^{-1} = \left(\sigma^2(t)I\right)^{-1} = \sigma^{-2}(t)I$, which yields the following simplification:

$$p_t(x|x_0) = \frac{1}{(2\pi)^{D/2}\sigma^D(t)}\exp\left(-\frac{\|x-\alpha(t)x_0\|^2}{2\sigma^2(t)}\right) \tag{17}$$

$$\log p_t(x|x_0) = -\frac{D}{2}\log(2\pi) - D\log\sigma(t) - \frac{\|x-\alpha(t)x_0\|^2}{2\sigma^2(t)} \tag{18}$$

$$\frac{\partial}{\partial t}\log p_t(x|x_0) = -D\frac{\dot\sigma(t)}{\sigma(t)} - \left[-\frac{\dot\sigma(t)}{\sigma^3(t)}\|x-\alpha(t)x_0\|^2 - \frac{1}{2\sigma^2(t)}\left(-2\dot\alpha(t)\left(x-\alpha(t)x_0\right)^\top x_0\right)\right]$$

$$= -D\frac{\dot\sigma(t)}{\sigma(t)} + \frac{\dot\sigma(t)}{\sigma^3(t)}\|x-\alpha(t)x_0\|^2 + \frac{\dot\alpha(t)}{\sigma^2(t)}\left(x-\alpha(t)x_0\right)^\top x_0$$

$$= -D\frac{\dot\sigma(t)}{\sigma(t)} + \frac{\dot\sigma(t)}{\sigma^3(t)}\sigma^2(t)\|z\|^2 + \frac{\dot\alpha(t)}{\sigma^2(t)}\left(\sigma(t)z\right)^\top x_0$$

$$= -D\frac{\dot\sigma(t)}{\sigma(t)} + \frac{\dot\sigma(t)}{\sigma(t)}\|z\|^2 + \frac{\dot\alpha(t)}{\sigma(t)}z^\top x_0$$

$$= \frac{\dot\sigma(t)}{\sigma(t)}(\|z\|^2 - D) + \frac{\dot\alpha(t)}{\sigma(t)}z^\top x_0 \tag{19}$$

Thus, $\mathcal{I}(t)$ can be decomposed into three components:

$$
\mathcal{I}(t) = \mathbb{E}_{x_0 \sim p_{\text{data}}} \mathbb{E}_{p_t(x|x_0)} \left[ \left( \frac{\dot{\sigma}(t)}{\sigma(t)} (\|z\|^2 - D) + \frac{\dot{\alpha}(t)}{\sigma(t)} z^\top x_0 \right)^2 \right]
$$

$$
= \mathbb{E}_{x_0 \sim p_{\text{data}}} \mathbb{E}_z \left[ \left( \frac{\dot{\sigma}(t)}{\sigma(t)} \right)^2 (\|z\|^2 - D)^2 + \left( \frac{\dot{\alpha}(t)}{\sigma(t)} \right)^2 (z^\top x_0)^2 + 2 \cdot \frac{\dot{\sigma}(t)\dot{\alpha}(t)}{\sigma^2(t)} (\|z\|^2 - D)(z^\top x_0) \right]
$$

$$
\tag{20}
$$

Since the first term follows a chi-squared distribution $\|z\|^2 = \sum_i z_i^2 \sim \chi^2(D)$, in this part, we introduce the data dimension $D$: $\mathbb{E}[\|z\|^2] = D$, $\mathbb{E}\left[(\|z\|^2 - D)^2\right] = \text{Var}[\|z\|^2] = 2D$

the second term is $\mathbb{E}\left[(z^\top x_0)^2\right] = \mathbb{E}\left[ \left(\sum_i z_i (x_0)_i\right) \left(\sum_j z_j (x_0)_j\right) \right] = \sum_{i,j} (x_0)_i (x_0)_j \delta_{ij} = \|x_0\|^2$

The third term, namely the cross-term, vanishes: $\mathbb{E}\left[(\|z\|^2 - D)(z^\top x_0)\right] = \mathbb{E}\left[\|z\|^2 \cdot (z^\top x_0)\right] - D \cdot \mathbb{E}[z^\top x_0] = \mathbb{E}\left[(\sum_i z_i^2)(\sum_j z_j (x_0)_j)\right] = \sum_{i,j} (x_0)_j \mathbb{E}[z_i^2 z_j] = 0$

$$
\mathcal{I}(t) = \mathbb{E}_{x_0 \sim p_{\text{data}}} \left[ \frac{\dot{\sigma}(t)}{\sigma(t)} \cdot 2D + \frac{\dot{\alpha}(t)}{\sigma(t)} \cdot \|x_0\|^2 \right]
\tag{21}
$$

In most prior works, due to the effect of data normalization, we can assume that $\mathbb{E}[x_0] = 0$, and therefore $\|x_0\|^2$ can be expressed in terms of $\text{Var}[x_0]$.

In the EDM framework, $\alpha(t) = 1$, $\sigma(t) = \sigma_{\text{data}} \cdot \left( s_{\max}^{1/p} + (1-t) \cdot (s_{\min}^{1/p} - s_{\max}^{1/p}) \right)^p$. This yields a more concise expression for $I(t)$:

$$
\mathcal{I}(t) = \frac{\dot{\sigma}(t)}{\sigma(t)} \cdot 2D = \frac{2D \cdot p \cdot (s_{\max}^{1/p} - s_{\min}^{1/p})}{s_{\max}^{1/p} + (1-t)(s_{\min}^{1/p} - s_{\max}^{1/p})}
\tag{22}
$$

## B  IMPLEMENTATION DETAILS

### B.1  TRAINING CONFIGURATION

To ensure clarity and reproducibility, we provide a detailed description of the training setup. Our full training pipeline was executed on a cluster equipped with 64 NVIDIA H800 GPUs, corresponding to an effective batch size of 64. Stage 1 focuses on learning diffusion consistency. DCFold was trained for approximately 40 hours, spanning a total of 9,000 optimization steps. This stage establishes the foundational generative capabilities leveraged in subsequent training. Stage 2 aims to refine the structural reasoning components through Pairformer consistency training. This phase required around 7 hours of computation and was conducted for 1,500 steps. The shorter duration reflects both the stability provided by Stage 1 and the efficiency of fine-tuning the Pairformer module.

### B.2  BINDER HALLUCINATION

After initial binder design with DCFold, sequences are refined to improve stability and solubility using ProteinMPNN with soluble weights, while preserving residues within 4 Å of the target interface. For each binder, 20 variants are generated at temperature 0.1 with no backbone noise. These sequences are re-predicted using the AF2 monomer model (3 recycles, 2 template-based models) in single-sequence mode to validate structural robustness. Resulting complexes are energy-minimized with Rosetta FastRelax (200 iterations) and evaluated using InterfaceAnalyzer with sidechain and backbone movement. Final designs are filtered using predefined thresholds (pLDDT $> 0.8$, i_pTM $> 0.5$, i_pAE $< 0.35$, shape complementarity $> 0.55$, $< 3$ unsaturated H-bonds, binder surface hydrophobicity $< 35\%$, RMSD $< 3.5$ Å), yielding a high-confidence set of candidates.

We evaluate binder quality using two constraint sets. Model-based Constraints are derived from AlphaFold2 confidence outputs, requiring pLDDT $> 0.8$, interface pTM $> 0.5$, global pTM $> 0.45$, and interface pAE $< 0.4$. Physics-based Constraints are based on physical interface metrics from Rosetta, including shape complementarity $> 0.5$, dSASA $> 1$, $> 6$ interface residues, $> 2$ interface hydrogen bonds, surface hydrophobicity $< 0.37$, and $< 6$ unsaturated hydrogen bonds. All metrics are aligned with the filters used in BindCraft.

Table 7: Average inference time of AlphaFold3 and DCFold across token bins.

| #Tokens | AlphaFold3 Avg Time (s) | DCFold Avg Time (s) |
|---|---|---|
| $\leq 255$ | 92.63 | 3.76 |
| 256–383 | 103.31 | 5.77 |
| 384–511 | 112.35 | 7.17 |
| 512–639 | 126.41 | 10.87 |
| 640–767 | 142.78 | 14.65 |
| 768–895 | 169.20 | 20.02 |
| $\geq 896$ | 212.12 | 27.40 |

### B.3 HYPERPARAMETER SETTINGS FOR CONSISTENCY MODEL BASELINES

For completeness, we provide the implementation details of all baselines considered in our experiments:

- CD: Mean squared error (MSE) as the metric function with a weight decay rate of $\eta = 0.995$.
- sCM: $H = 2000$ warm-up iterations.
- ECM: $q = 2.0$, $b = 0.1$, $d = 3000$, and $k = 4.0$.
- TGM: Hyperparameter search yields $C_0 = 32$ and $\beta = 2$. In addition, we inherit the exponential decay scheduling parameters from AlphaFold3's EDM configuration, with $p = 7$, $s_{\max} = 160$, and $s_{\min} = 4 \times 10^{-4}$.

For all methods, we set the weighting function to 1.

## C EXPERIMENT DETAILS

### C.1 RUNTIME CHARACTERISTICS ACROSS SEQUENCE LENGTHS

To comprehensively assess the efficiency of DCFold, we report detailed bin-wise runtime statistics on the Posebusters V2 benchmark. Since AlphaFold3 supports folding protein-ligand complexes, we use the total number of input tokens for each test entry as the length metric and partition sequences into bins of size 128. The average inference time for each bin is summarized in Table 7.

Both AlphaFold3 and DCFold exhibit increasing runtime as sequence length grows. However, the relative acceleration provided by DCFold is most pronounced for short sequences, where it achieves up to a **24×** speedup. For moderately long sequences, DCFold still provides more than **7.7×** acceleration, demonstrating consistent efficiency gains across all token ranges.

We hypothesize that this trend stems from the differing computational bottlenecks of the two methods. The reduction in Diffusion NFE afforded by DCFold yields a significantly larger improvement factor compared to the reduction in Pairformer cycles. As sequence length increases, the Pairformer component becomes the dominant cost, diminishing the relative impact of the diffusion speedup. Conversely, in shorter sequences, the Pairformer bottleneck is less pronounced, enabling the diffusion efficiency gains to translate directly into substantial end-to-end acceleration.

### C.2 BINDER HALLUCINATION

We conducted experiments on a single H800 GPU. On the targets used in Table 5, the average GPU time for one full hallucination with BindCraft is 138s, while DCFold requires 105s. Since we follow the same pipeline as BindCraft, the total serial runtime also includes the time for ProteinMPNN and the re-prediction step in addition to the design model's GPU time. We also provide the total number of designs generated in our experiments in Table 8. Overall, DCFold attains slightly better efficiency while producing a comparable number of samples, ensuring a fair comparison.

Our binder design benchmark features six protein targets. Table 9 shows the details of the targets.

### THE USE OF LARGE LANGUAGE MODELS (LLMS)

We use large language models (LLMs) solely for auxiliary editing purposes, including spelling correction and minor grammatical adjustments. Importantly, LLMs are not involved in the conception

Table 8: The total number of generated samples in the binder hallucination experiments.

| | IL-2R$\alpha$ | TrkA | H3 | VirB8 | ALK | LTK |
|---|---|---|---|---|---|---|
| BindCraft | 312 | 243 | 269 | 347 | 188 | 348 |
| DCFold (Ours) | 375 | 256 | 295 | 439 | 177 | 402 |

Table 9: Detailed information of binder targets in the binder hallucination experiments.

| Target | PDB ID | Family | Description |
|---|---|---|---|
| ALK | 7NWZ | Immune receptor | Neural receptor tyrosine kinase involved in development |
| H3 | 3ZTJ | Receptor tyrosine kinase | Core nucleosomal histone in eukaryotic chromatin |
| IL2R$\alpha$ | 1Z92 | Histone protein | Component of the interleukin-2 receptor complex in the immune system |
| LTK | 7NX0 | Bacterial secretion system protein | Homolog of ALK expressed in various tissues |
| TrkA | 2IFG | Receptor tyrosine kinase | Neurotrophic signaling receptor activated by NGF |
| VirB8 | 4O3V | Receptor tyrosine kinase | Structural protein of the type IV secretion system in Gram-negative bacteria |

of research ideas or the development of code. We disclose this usage explicitly to ensure transparency in our work.

