# OpenReview forum: "DCFold: Efficient Protein Structure Generation with Single Forward Pass"
_ICLR.cc/2026/Conference — ICLR 2026 Oral_

### Official Review · Reviewer_M62B · 2025-10-16

**Soundness:** 2
**Presentation:** 1
**Contribution:** 2
**Rating:** 4
**Confidence:** 4

**Summary:**

The paper introduced DCFOLD, a distilled version of AlphaFold3 (AF3) that generates protein structures in a single forward pass, achieving up to 15× faster inference while maintaining similar accuracy. The proposed method introduces a training framework that: 1) aligns one step predictions with the full multi-step diffusion process and 2) enforces representation alignment on AF recycling iterations. The paper introduces a scheduler which aims to stabilize the training on variable-length proteins. Experiments on PoseBusters V2, Recent PDB, and binder hallucination tasks claim near AF3-level accuracy with reduced inference cost.

**Strengths:**

1. The proposed framework addresses a bottleneck in AF3 as it is computationally at inference time. Speeding up the inference time will help to address tasks such as protein design and virtual screening.

2. The DC framework is relatively novel.

3. The results are promising reporting a 15x speedup while retaining similar performances on selected tasks.

**Weaknesses:**

1. Several details are missing or inconsistently used. Some examples:
- In Eq. 1, $w(t)$ is not defined, Later, it is set to 1, but its role is never justified.
- The reference timestep $r$ is ambiguous in Algorithm 1. Is it a function or scalar?
- The variable $u \in [0, 1]$, in Algorithm 1 is introduced as "training progress" but never clearly explained.
- The stop gradient operator is defined in Eq. 2, but never used anywhere

2. Missing important details make the training procedure non-reproducible.
- $\mathcal{L}$ in Algorithm 1 seems to be a combination of $\mathcal{L}_{diffusion}$, $\mathcal{L}_{pairwise}$, and $\mathcal{L}_{confidence}$ (this latter never defined). The combination of those three losses is unclear. Are they combined linearly or trained sequentially in stages? If sequential, how are modules frozen or fine-tuned? What are the relative weighting coefficients?
- $\alpha$ defined after Eq. 3 is unclear. Sometimes is a scalar, sometimes a vector, sometimes a pairwise matrix?

3. The evaluation setup is significantly weaker than the Boltz2[1] benchmark used for AF3 and related models. Here, some of them:
- FEP accuracy on public benchmarks
- Local protein dynamics

4. The theoretical part about TGM lacks clarity. E.g. what are $C_0$, $s_{max}$, and $s_{min}$?

[1] Passaro, Saro, et al. "Boltz-2: Towards accurate and efficient binding affinity prediction." BioRxiv (2025): 2025-06.

**Questions:**

Please see in the Weakness section some of my questions. More details questions follow:

1. how is $u$ scheduled or computed in practice?
2. how is $r$ chosen in Algoritm 1?
3. how are categorical features handled?

Minor:
1. Text clarity can be improved
2. Some of the references need to be fixed e.g. AlphaFold3 line 306 page 6 is referenced to AlphaFold2 paper.

---

> ### Author Response · Authors · 2025-11-19
> **Response to Reviewer M62B (1/n)**
>
> # W1 & Q1 & Q2 Additional details on the definition and meaning of the notation
> We sincerely appreciate the points you raised regarding the clarity of our writing and provide detailed explanations below:
>
> The consistency model loss in Eq. 1 follows the formulation in [1], where $w: \mathbb{R}_{\ge 0} \to \mathbb{R}^+$ denotes a positive weighting function. Prior works on consistency models typically employ different loss weights for different timesteps. Our experiments indicate that TGM is not highly sensitive to the choice of this function; therefore, we simply set $w(\cdot) \equiv 1$ to simplify implementation. Nevertheless, within the TGM framework, this remains a flexible degree of freedom that can be adjusted for other applications.
>
> In Step 3 of Algorithm 1, our intention was to clip the computed value of $r$ to 0. A clearer formulation would be:
> 3: $r' \leftarrow \max(r(t,u), 0)$,
> 4: $x_{r'} \leftarrow x_0 + r'\epsilon$,
> 5: $\mathcal{L} \leftarrow w(t) \Vert f_\theta(x_t,t) - f_{\theta^-}(x_{r'},r) \Vert_2^2$.
> Here, $r(t,u)$ indicates that $r$ is a function of $t$ and $u$, while $r'$ is the scalar result after clipping. In our implementation, multiple samples are processed in parallel, so $r'$ and $t$ can also be treated as vectors, with element-wise operations in $r(t,u)$.
>
> The training progress variable $u$ in Algorithm 1 is inherited from ECM [2], defined as $u = \frac{steps}{total ~steps}$, representing the current fraction of total training steps completed. We expect that as training progresses, the gap between $t$ and $r(t,u)$ decreases, thereby enforcing stronger consistency in the model. This also explains the origin of the computation of $r(t,u)$ in Section 3.3.
>
> The stop-gradient operator in Eq. 2 is applied immediately to prevent gradients from earlier timesteps, enforcing alignment only from outputs at noisier timesteps to those at less noisy timesteps. This is a common practice in most previous consistency model works for stable training. In Algorithm 1, this process is also shown in Step 5. We acknowledge that the use of the inconsistent notation $\theta^-$ may have caused confusion.
>
> We sincerely apologize for the improper definitions and usage of various symbols in our manuscript. In the revision, we will correct all of them to improve readability. We hope these issues do not affect the evaluation of the novelty of our work, and we thank you for helping us improve it.
>
> -----
> # W2 Additional details on the training procedure
> We thank the reviewer for pointing out the unclear presentation in our manuscript, and we address each point as follows:
>
> The term $\mathcal{L}_{\text{confidence}}$ is inherited from the concept described in the AlphaFold3 Supplementary Materials, and is explicitly composed as:
>
> $L_{confidence} = \alpha_{confidence} \cdot (L_{plddt} + L_{pde} + L_{resolved} + \alpha_{pae} \cdot L_{pae})$
>
> where $\alpha_{\text{confidence}} = 10^{-4}$ and $\alpha_{\text{pae}} = 1$, fully following AlphaFold3's settings in the final training stage. The specific details of these losses can be referred to in AlphaFold3, and, in general, they are computed from the predicted confidence of the model in relation to the error between predicted and true structures. We adopt $\mathcal{L}_{\text{confidence}}$ in order to leverage the information from the frozen confidence module, ensuring that the model's internal features, which maintain consistency, can still produce reasonable confidence scores.
>
> In Table 1, we report which losses are enabled at each training stage. In Stage 1, the confidence and diffusion losses are enabled to train Diffusion Consistency; in Stage 2, the confidence and pairformer losses are enabled to train Pairformer Consistency. We clarify at the end of Section 3.2 that the weights are frozen: in Stage 1, only the diffusion module is updated while all other modules are frozen; in Stage 2, only the pairformer module is updated while all others remain frozen. What we omitted to mention in the main text is that enabling a loss means adding it to the total loss with a weight of 1, and the total loss is then used to compute gradients.
>
> Below Eq. 3, we define $\alpha$ as "the per-token weighting coefficient used in AlphaFold3," indicating that it is a vector with length equal to the number of sequence tokens. We would like to further clarify that $\alpha$ is a column vector; consequently, $\sqrt{\alpha} \sqrt{\alpha}^\top$ produces a pair-wise matrix, which is then used to weight the pair feature $z$ with values in the same range as $\alpha$.
>
> We sincerely apologize for the oversight in our writing and will revise these sections in the updated version to improve clarity. We thank the reviewer for their valuable contribution to our work.

---

> ### Author Response · Authors · 2025-11-19
> **Response to Reviewer M62B (2/n)**
>
> # W3 Additional benchmarks related to affinity
> We appreciate the reviewer highlighting Boltz2’s focus on important tasks such as local protein dynamics and affinity prediction, which generalize beyond a conventional folding model. Indeed, these are highly relevant problems in small-molecule and binder design. However, this is exactly what makes the benchmarks used in Boltz2 not directly applicable to our current evaluation:
>
> 1. According to Section 5.2 of [3], Boltz2's success in local protein dynamics experiments stems from its training on a large number of MD ensembles, with “supervision on both experimental and computational B-factor estimates.” This indicates that this task cannot be adequately addressed by a conventional folding model; training data would need to be replaced to achieve good performance on these benchmarks. Moreover, as we discussed in our response to reviewer 9xiz, neither AlphaFold3 nor DCFold tends to generate significantly diverse conformations. Therefore, we consider this benchmark unsuitable for our task.
> 2. Large-scale virtual screening and affinity prediction (FEP accuracy on public benchmarks) rely on Boltz2's newly introduced affinity module, which runs in parallel with the confidence module. Since AlphaFold3 and DCFold do not have a corresponding module, these experiments cannot be conducted under our current experimental setup.
>
> The practicality of DCFold, however, is demonstrated by its success in binder design, where it achieves both high success rates and substantial efficiency gains. This showcases its ability to unlock potential speedups for downstream applications, including those targeted by Boltz2.
>
> Importantly, the good thing is that the proposed Dual Consistency approach is agnostic to a specific AlphaFold3 implementation. It can be directly applied to any model containing both a Pairformer and a diffusion module, including Boltz2. Integrating Dual Consistency into Boltz2 would immediately provide a high-speed, high-accuracy variant for these benchmarks. Exploring this integration is an exciting direction for future work, but the required effort exceeds the scope of the current study.
>
> -----
> # W4 Additional details on the symbol definitions in TGM
> We appreciate your pointing out the unclear parts in our writing, and we address them point by point below:
>
> In Section 3.3, we use $C(u)$ to control the smooth decay of $d_g(t,r)$ over the course of training. Theoretically, $C(u)$ can be any monotonically decreasing function. In our experiments, we chose a parameterization that yielded the best performance, namely $C(0) = C_0$, $C(1) = 0$, and $C(u) = C_0(1-u)^\beta$ with $\beta>0$. Therefore, $C_0$ can be regarded as a hyperparameter in DCFold training that controls the geodesic distance between two time points at the beginning of training. In Appendix B.2, we specify that we used $C_0=32$ to enhance the reproducibility of our method.
>
> The parameters $s_{\text{max}}$ and $s_{\text{min}}$ follow the definitions in EDM [4], which provides a unified generative modeling framework and has been adopted in AlphaFold3’s diffusion module. $s_{\text{max}}$ and $s_{\text{min}}$ control the maximum and minimum noise strengths during inference, respectively. The supplementary materials of AlphaFold3 report the hyperparameters as $s_{\text{max}}=160$ and $s_{\text{min}}=4\cdot 10^{-4}$. During a 200-step inference, the noise strength gradually decays from $\sigma_{\text{data}} \cdot s_{\text{max}}$ to $\sigma_{\text{data}} \cdot s_{\text{min}}$, where $\sigma_{\text{data}}$ denotes the variance of the dataset coordinates and is set to 16.
>
> We acknowledge that these concepts were not clearly identified in the main text. We sincerely apologize and will correct all ambiguities in the revision. We thank you for your contribution to improving the readability of our paper!

---

> ### Author Response · Authors · 2025-11-19
> **Response to Reviewer M62B (3/3)**
>
> # Q3 Additional details on the processing methods for categorical features
> Thank you for raising this question. The "categorical features" mentioned in your comment could refer to multiple aspects, and we address each possibility as follows:
>
> 1. If you are referring to the input categorical features in DCFold, we have made no modifications to this part of the algorithm and fully inherit the approach from AlphaFold3. In AlphaFold3, categorical features are generally converted to one-hot encodings, such as residue types, elements in the MSA, and elements in the reference conformer, allowing them to be treated as vectors in a continuous space.
>
> 2. If you are referring to categorical features in the diffusion process, it is important to note that neither AlphaFold3 nor DCFold involves categorical features in their diffusion process. The diffusion process starts from pure noise in 3D coordinates and progressively denoises it to produce predicted full-atom coordinates, where all variables are continuous.
>
> However, considering how to apply Dual Consistency, especially TGM, to categorical features (i.e., discrete diffusion) is an interesting direction. Our TGM framework does not rely on a continuous modality, as it focuses on determining the reference timestep corresponding to $t$ during consistency training, which is a critical and consistently implemented component in both continuous and discrete consistency models. The main task would be to rewrite $\mathcal{I}(t)$ using the parameterization suitable for discrete diffusion. Yet, prior works have observed that this can lead to difficulties for the student model in fitting the teacher's discrete outputs. Making it fully practical thus requires further investigation, which we leave as future work.
>
> -----
> # Q4 & Q5 Clarity of writing and corrections to citations
> Thank you for pointing out the error! We will carefully consider all reviewers' feedback and aim to present the most accurate and clear descriptions in the revision. We sincerely appreciate your correction!
>
> [1] Song, Yang, et al. "Consistency models." (2023).
>
> [2] Geng, Zhengyang, et al. "Consistency models made easy." arXiv preprint arXiv:2406.14548 (2024).
>
> [3] Passaro, Saro, et al. "Boltz-2: Towards accurate and efficient binding affinity prediction." BioRxiv (2025): 2025-06.
>
> [4] Karras, Tero, et al. "Elucidating the design space of diffusion-based generative models." Advances in neural information processing systems 35 (2022): 26565-26577.

---

> ### Comment · Reviewer_M62B · 2025-11-28
> **Response**
>
> Thanks! All my concerns have been resolved. I am happy to increase my score to 6

---

### Official Review · Reviewer_KePm · 2025-10-29

**Soundness:** 3
**Presentation:** 3
**Contribution:** 3
**Rating:** 4
**Confidence:** 3

**Summary:**

This paper introduces **DCFold**, a single-step generative model for protein structure prediction, designed to overcome the slow inference speed of the iterative, diffusion-based AlphaFold3.

The core contribution is a **Dual Consistency** training framework, featuring a novel **Temporal Geodesic Matching (TGM) scheduler**. This allows DCFold to achieve AlphaFold3-level accuracy in a single generative step.

The authors report a **15x acceleration in inference time** while maintaining high predictive fidelity, with performance validated on both structure prediction and binder design tasks.

**Strengths:**

1.  Good acceleration was achieved by simultaneously distilling the Pairformer and the diffusion model, with no obvious degradation in performance.
2.  The paper proposes a novel time step sampling strategy that effectively stabilizes training and improves the model's performance.
3.  The mathematical derivations in the appendix are easy to follow and have no obvious errors.

**Weaknesses:**

1.  **Lack of Granularity in Inference Time Analysis:** The time complexity of different computational stages in protein structure prediction scales differently with protein length (L). As such, the inference time for a model like AlphaFold3 can vary significantly across proteins of different lengths. The manuscript only reports the mean inference time on the evaluation set, which obscures these details. A more thorough analysis showing a direct comparison of inference times for DCFold and AlphaFold3 binned by protein length would be necessary to fully assess the claimed acceleration and understand its behavior in different regimes.

2.  **Ambiguity in Pose Selection for PoseBusters Benchmark:** The methodology for selecting the "best" and "worst" poses in the PoseBusters benchmark results is unclear. If the ground truth structure was used to rank the generated poses and select these examples, this would constitute a form of data leakage, as the ground truth is not available in a real-world prediction scenario. Since ranking generated poses without a ground truth is a significant challenge in itself, the authors should clarify this selection process to ensure the validity of the reported results.

3.  **Unclear Metric Definition and Counter-intuitive Results in Table 3:** The definition of "Success Rate" used in *Table 3* is not clearly described in the manuscript. Furthermore, the results presented are somewhat surprising: the distilled, single-step model reportedly outperforms the original AlphaFold3 teacher model across the board on the Homology Recent PDB dataset. While not impossible, it is counter-intuitive for a student model to uniformly exceed the performance of its teacher. This warrants a more detailed explanation of the metric and a discussion of these results.

4.  **Insufficient Detail in Binder Hallucination Throughput:** The evaluation of the binder hallucination task lacks sufficient detail to assess the claimed speed benefits. The paper states that a "continuous 48-hour hallucination run" was performed and reports the *proportion* of designs that passed filters. However, it omits the crucial information of the *total number* of designs generated by DCFold and the baseline (BindCraft) within that time frame. Without this denominator, it is impossible to quantify the throughput advantage gained from the distillation process.

5.  **Unconventional Experimental Setup and Incomplete Results for Binder Design:** The choice of a 55-65 residue length for the binder design task is an unconventional setting that requires justification, as this is a relatively short range. More importantly, the evaluation is limited to quantitative metrics. The manuscript does not show any examples of the successfully designed binder structures. This absence of qualitative, visual evidence makes it difficult to independently assess the structural plausibility and quality of the generated binders.

**Questions:**

1.  **Regarding the Gradient-Based Hallucination Method:** Could the authors provide more detail on the implementation of the gradient-based hallucination? The iterative nature of diffusion models makes obtaining a direct gradient with respect to the final structure non-trivial. Methods like BoltzDesign circumvent this by using gradients from the Pairformer's internal representations. Does DCFold's single-step nature allow for a direct backpropagation through the structure generation process, similar to approaches like BindCraft, or is another mechanism used?

2.  **Stochasticity and Sampling Diversity in a Single-Step Model:** The SDE-based sampling in diffusion models like AlphaFold3 provides a beneficial stochasticity, which can capture protein dynamics and yield improved results when multiple samples are drawn. Does the single-step, distilled architecture of DCFold retain this inherent randomness? Furthermore, can DCFold's performance be improved by generating multiple samples for a single target, and if so, could the authors show the trend of performance improvement as the number of samples increases?

---

> ### Author Response · Authors · 2025-11-19
> **Response to Reviewer KePm (1/n)**
>
> # W1 The relationship between time improvement and sequence length
> Thank you for raising this point; it is crucial for a more detailed investigation of DCFold's performance. We provide more detailed bin-wise runtime statistics on Posebusters V2 in Figure 1. Since AlphaFold3 can fold complexes containing small molecules, we use the number of input tokens as the length of each test entry and partition bins in units of 128. On both AlphaFold3 and DCFold, runtime increases with sequence length. Our method, DCFold, exhibits more pronounced speedup on shorter sequences, achieving up to a 24× acceleration. For moderately long sequences, DCFold still attains over 7.7× speedup.
>
> We hypothesize that the improvement factor in Diffusion NFE is significantly larger than that in the number of Pairformer cycles. Consequently, for long-sequence folding, overall runtime is more constrained by the Pairformer bottleneck, whereas for shorter sequences, the Pairformer bottleneck is less pronounced, allowing for greater acceleration.
>
> This order-of-magnitude speedup is highly beneficial for downstream tasks. Many structure design tasks require repeatedly generating large numbers of samples to assess feasibility or increase diversity, and the reduction in inference time linearly increases the number of samples that can be generated. The up to 24× acceleration on short sequences makes the previously time-consuming sampling process substantially lighter, enabling researchers to explore larger structural spaces under the same computational budget, thereby improving design quality and success rates.
>
> | #Tokens     | AlphaFold3 Avg Time (s) | DCFold Avg Time (s) |
> |-------------|--------------------------|-----------------------|
> | <255        | 92.63                    | 3.76                  |
> | 256–383     | 103.31                   | 5.77                  |
> | 384–511     | 112.35                   | 7.17                  |
> | 512–639     | 126.41                   | 10.87                 |
> | 640–767     | 142.78                   | 14.65                 |
> | 768–895     | 169.20                   | 20.02                 |
> | >896        | 212.12                   | 27.40                 |
>
> -----
> # W2 The method for computing the best and worst samples
> We thank the reviewer for bringing this up and helping us improve the clarity of our manuscript. As the reviewer correctly understood, the values in Table 2 are computed with respect to the ground-truth structures. However, this does not constitute any form of data leakage; instead, it is simply used to measure the range of the minimum and maximum $\text{RMSD}$ achieved by the sampled structures.
>
> To clarify the evaluation protocol: in Section 4.1, DCFold generates 5 samples for each test entry under the same random seed, computes the $\text{RMSD}$ to the ground-truth structure for each sample, and ranks these samples accordingly. The lowest $\text{RMSD}$ is reported as the *best*, and the highest as the *worst*. Because Table 2 aims to compare the **distributional differences** between AlphaFold3’s and DCFold’s predictions—rather than to assess the **correlation between confidence and structural quality**—using ground truth here does not leak information into the model. Our intention is to show that DCFold exhibits a higher upper bound on sample quality and a higher lower bound compared to AlphaFold3, demonstrating DCFold’s ability to avoid low-quality predictions.
>
> Regarding confidence prediction, DCFold inherits AlphaFold3’s performance because the confidence module is frozen during training. As shown in our response to reviewer 9xiz - Question 1, a direct comparison between AlphaFold3 and DCFold confidence scores confirms that DCFold maintains AlphaFold3’s capability in confidence estimation.

---

> ### Author Response · Authors · 2025-11-19
> **Response to Reviewer KePm (2/n)**
>
> # W3 Definition of the success rate and why DCFold outperforms AlphaFold3 on TM-score and SR metric
> We sincerely apologize for any confusion caused by our presentation. The Success Rate in Table 3 follows the RMSD Success Rate in Protenix[1], which is defined as the percentage of predictions with $RMSD < 2\text{A}$. When $RMSD < 2\text{A}$, the predicted structure reproduces the key binding geometry, preserves important side chains and the pocket, and provides a reliable binding pose. Therefore, $2\text{A}$ is a reasonable threshold for defining success. We will clarify this definition in the revision.
>
> Regarding the reviewer’s question about DCFold surpassing AlphaFold3 in Table 3, we believe this “anomalous” phenomenon is meaningful and worth discussing. Our perspectives are as follows:
>
> 1. In the field of image generation, prior work has observed that, in a small fraction of cases, a fine-tuned model can indeed surpass its pretrained counterpart, e.g., ECM[2] Table 5. Maintaining model consistency can even improve the performance of the pretrained model, making it possible for a student model to exceed the teacher. Unlike simple distillation, consistency training imposes a stricter constraint than diffusion training, which corresponds to the most relaxed consistency condition ($\Delta t = T$). Thus, a consistency model can be viewed as a form of post-training that enforces a tighter mapping along the denoising trajectory. During this process, the model may acquire new capabilities and produce a stronger student.
> 2. AlphaFold3’s generative process differs from many other generative models in an important way: its stochastic diffusion trajectory is not always the most efficient path toward the underlying structural mode. Stochastic sampling may introduce unnecessary noise or deviations at intermediate steps. In contrast, DCFold leverages Dual Consistency to progressively learn a direct mapping from noisy inputs to clean structures. Moreover, the practice in BoltzDesign[3] of computing the loss and backpropagating using Pairformer-predicted distograms also indicates that most structural information is already determined before the diffusion step, supports our approach of learning a direct mapping.
> 3. According to our response to reviewer 9xiz — Question 1, we confirm that DCFold has only a minor impact on diversity while maintaining high confidence, indicating that DCFold preserves the mapping to high-quality and diverse structures.
>
> We appreciate the reviewer’s careful observations, which allow us to more clearly articulate the fundamental reasons behind DCFold’s performance. We will incorporate these discussions in the revision.
>
> -----
> # W4 Details of the binder design experiment
> Thank you for your valuable suggestions regarding our experimental section. We conducted experiments on a single H800 GPU. On the targets used in Table 4, the average GPU time for one full hallucination with BindCraft is 138s, while DCFold requires 105s. Since we follow the same pipeline as BindCraft, the total serial runtime also includes the time for ProteinMPNN and the re-prediction step in addition to the design model’s GPU time. We also provide the total number of designs generated in our experiments in the table below. Overall, DCFold attains slightly better efficiency while producing a comparable number of samples, ensuring a fair comparison.
>
> As mentioned in our responses to Reviewer XzaU – Question 1 and Weakness 1 regarding the relationship between sequence length and runtime, DCFold’s performance gains are even more pronounced on simpler targets (e.g., VirB8).
>
> | Target        | ALK(7NWZ) | H3(3ZTJ) | IL2Ra(1Z92) | LTK(7NX0) | TrkA(2IFG) | VirB8(4O3V) |
> |---------------|------------|-----------|--------------|------------|-------------|--------------|
> | **BindCraft**  | 188        | 269       | 312          | 348        | 243         | 347          |
> | **DCFold**     | 177        | 295       | 375          | 402        | 256         | 439          |

---

> ### Author Response · Authors · 2025-11-19
> **Response to Reviewer KePm (3/n)**
>
> # W5 Selection of binder sequence length and visualization of binder design
> We appreciate the reviewer’s question regarding the details of our work. We acknowledge that the residue lengths of 55–65 are indeed shorter compared to BindCraft[4] and the corresponding experimental crystal structures of the targets in the PDB. Our choice of this length range is motivated by two considerations:
>
> 1. **Consistency with the standard benchmark setting for de novo binder design.**
>    A large body of prior work[5,6] has established mini-proteins in the 50–70 residue range as the standard benchmark for de novo binder design. Our evaluation targets are taken directly from[6], and following this established length regime ensures comparability with widely recognized baselines and prior experimental protocols.
> 2. **Strong biological grounding of the mini-protein regime.**
>    The 55–65 residue range is not merely a computational choice. Mini-proteins of this size have repeatedly been shown to fold stably, adopt well-defined rigid interfaces, and support high-affinity binding in experimental settings[6]. This length regime naturally accommodates multiple secondary-structure elements (helices, loops) and diverse binding topologies, making it biologically meaningful and widely adopted in de novo binder design studies
>
> -----
> # Q1 DCFold allows hallucination starting from a structure
> As you pointed out in your question, our work is precisely a structure-first hallucination approach. This is precisely the motivation behind DCFold. Unlike BoltzDesign[3], which starts from Pairformer-predicted distograms and optimizes sequences via gradients, avoiding the computational overhead and gradient instability of backpropagating through diffusion, we perform hallucination starting from the AlphaFold3 structure. By directly propagating gradients from the structure, DCFold enjoys the benefit of a better resolution to leverage AlphaFold3’s knowledge and can achieve superior results. To make this practical, we employ Dual Consistency to drastically reduce the number of cycles between the Pairformer and the Diffusion Module, effectively compressing network depth and significantly lowering the difficulty of gradient backpropagation.
>
> -----
> # Q2 The impact of consistency models and multi-step sampling on diversity
> Thank you for your insightful questions, which helped us further investigate the performance advantages of DCFold. We have added the Diversity and Confidence metrics for both AlphaFold3 and DCFold on Posebusters V2, and found that there is no significant shift in terms of diversity and confidence. For each sequence, we sample 5 structures and compute the pairwise TM-score among these structures; the dataset-level average serves as the Diversity metric (lower is better). We compute the LDDT of all sampled structures as the Confidence metric (higher is better).
>
> Our experiments show that, after Dual Consistency training, DCFold exhibits no substantial changes in these metrics compared with AlphaFold3: diversity becomes slightly worse while confidence increases slightly. This indicates that the process of aligning representations/structures in Dual Consistency tightens the structure distribution to some extent while preserving high-quality predictions. This observation is consistent with our statement in Section 4.1 that “such a redistribution reduces extreme errors ...”.
>
> We further conducted experiments by sampling more structures under the same random seed and sampling under different seeds. We observed that neither AlphaFold3 nor DCFold exhibits noticeable improvements in diversity. In our response to Reviewer KePm’s Weakness 3, we discussed the strong conditionality of the AlphaFold family of models, which we believe is closely connected to the difficulty of improving their sampling diversity. It is worth noting, however, that due to the intrinsic conformational heterogeneity of proteins, many prior works have focused on enhancing the sampling diversity of AlphaFold-like models, including but not limited to sampling MSAs, clustering and masking MSA columns, and tuning dropout rates [7–9]. Our acceleration method is orthogonal to most existing diversity-enhancing strategies. These prior techniques can also be applied to DCFold, and we expect them to yield similar improvements as those previously observed on AlphaFold3. We again thank you for raising these questions, which enabled a deeper discussion of this topic.
>
> | Method   | Diversity (↓)       | Confidence (↑)      |
> |----------|---------------------|---------------------|
> | AF3      | 0.9646 ± 0.0410     | 93.97 ± 2.92        |
> | AF3 (15 samples)     | 0.9642 ± 0.0415        | 93.95 ± 2.93           |
> | AF3 (5 seeds)        | 0.9697 ± 0.0421        | 93.90 ± 3.01           |
> | DCFold   | 0.9701 ± 0.0565     | 94.14 ± 2.97        |
> | DCFold (15 samples)  | 0.9708 ± 0.0567        | 94.13 ± 2.96           |
> | DCFold (5 seeds)     | 0.9712 ± 0.0570        | 94.15 ± 2.97           |

---

> ### Author Response · Authors · 2025-11-19
> **Response to Reviewer KePm (4/4)**
>
> [1] Team, ByteDance AML AI4Science. "Protenix." (2024).
>
> [2] Geng, Zhengyang, et al. "Consistency models made easy." arXiv preprint arXiv:2406.14548 (2024).
>
> [3] Cho, Yehlin, et al. "Boltzdesign1: Inverting all-atom structure prediction model for generalized biomolecular binder design." bioRxiv (2025): 2025-04.
>
> [4] Pacesa, Martin, et al. "BindCraft: one-shot design of functional protein binders." bioRxiv (2024): 2024-09.
>
> [5] Chevalier, Aaron, et al. "Massively parallel de novo protein design for targeted therapeutics." Nature 550.7674 (2017): 74-79.
>
> [6] Cao, Longxing, et al. "Design of protein-binding proteins from the target structure alone." Nature 605.7910 (2022): 551-560.
>
> [7] Wayment-Steele, Hannah K., et al. "Predicting multiple conformations via sequence clustering and AlphaFold2." Nature 625.7996 (2024): 832-839.
>
> [8] Wallner, Björn. "AFsample: improving multimer prediction with AlphaFold using massive sampling." Bioinformatics 39.9 (2023): btad573.
>
> [9] Kalakoti, Yogesh, and Björn Wallner. "AFsample2 predicts multiple conformations and ensembles with AlphaFold2." Communications Biology 8.1 (2025): 373.

---

### Official Review · Reviewer_XzaU · 2025-10-29

**Soundness:** 3
**Presentation:** 4
**Contribution:** 4
**Rating:** 8
**Confidence:** 4

**Summary:**

The paper presents DCFold, a model that greatly increases inference speed for AlphaFold3, which is very important for downstream tasks, such as virtual screening. It achieves this performance using an implementation of consistency models on both the model trunk, and the diffusion head. The authors also add a "Temporal Geodesic Matching" scheduler to stabilize consistency training. The model achieves the promised inference speed-up without sacrificing performance

**Strengths:**

**Originality** There have been other attempts to speed up implementations of AlphaFold3. However, the idea of using consistency training for both the trunk and the diffusion is very innovative, as well as the addition of Temporal Geodesic Matching.

**Quality**: The experiments are thorough and convincing, and the results show a good model performance

**Clarity**: The manuscript is very well written, and clearly explained. The figures are generally good, particularly figure 2, which is a very clear illustration of the approach.

**Significance**: As the authors explain, speeding up inference for AlphaFold3 is very important for multiple downstream applications like virtual screening

**Weaknesses:**

There are no major weaknesses, just some small things:

- The meaning of ECM is never introduced

**Questions:**

- For Alphafold, inference time is greatly dependent on sequence length. Is this 15x time improvement consistent across lengths?
- How did the training time compare with the training time of the original AlphaFold code?
- The benchmarks in table 2 only compare performance against AlphaFold3. If I understand correctly, this is referring to the protenix implementation of AlphaFold3 (which never achieved the same performance). This should be clarified. If these numbers are indeed from protenix, it would also be interesting to see a comparison against other implementations (the original AlphaFold3, Boltz2, OpenFold3, etc)
- It would also be interesting to see a comparison with protenix-mini, another attempt to reduce inference time (with a different approach) as well as a discussion about the difference between the two codes.
- The meaning of ECM is never introduced
- How is "in silico success rate" defined in Table 4?
- In the same table, results from other binder design algorithms (e.g. RFDiffusion, FoldFlow) should also be added.

---

> ### Author Response · Authors · 2025-11-19
> **Response to Reviewer XzaU (1/n)**
>
> # W1 & Q5 Introduction to ECM
> We sincerely apologize for our oversight in the baseline description. ECM (Consistency Models trained with Easy Consistency Tuning) [1], published at ICLR 2025, proposes a new scheduler
>
> $\frac{r}{t} = 1-\frac{1}{q\lfloor \text{iters}/d \rfloor}\left(1+\frac{k}{1+\exp(bt)}\right)$,
>
> where $q, d, k, b$ are hyperparameters. This scheduler simplifies and accelerates consistency model training, achieving state-of-the-art performance among few-step diffusion models on CIFAR-10 and ImageNet. In Section 4.3 and Table 5, we apply its method to AlphaFold3 distillation and observe strong empirical performance. However, due to its large number of hyperparameters and high sensitivity to their configuration, ECM does not support fine-grained tuning under the computationally expensive AlphaFold3 setting, and—like prior approaches—cannot adapt to changes in sequence length. Therefore, we select ECM as a strong baseline and show that TGM achieves comparable training efficiency with superior generation quality.
>
> We will include its description in the revision version. We appreciate your feedback, which helps improve the clarity of our work.
>
> -----
> # Q1 The relationship between time improvement and sequence length
> Thank you for raising this point; it is crucial for a more detailed investigation of DCFold's performance. We provide more detailed bin-wise runtime statistics on Posebusters V2 in Figure 1. Since AlphaFold3 can fold complexes containing small molecules, we use the number of input tokens as the length of each test entry and partition bins in units of 128. On both AlphaFold3 and DCFold, runtime increases with sequence length. Our method, DCFold, exhibits more pronounced speedup on shorter sequences, achieving up to a 24× acceleration. For moderately long sequences, DCFold still attains over 7.7× speedup.
>
> We hypothesize that the improvement factor in Diffusion NFE is significantly larger than that in the number of Pairformer cycles. Consequently, for long-sequence folding, overall runtime is more constrained by the Pairformer bottleneck, whereas for shorter sequences, the Pairformer bottleneck is less pronounced, allowing for greater acceleration.
>
> This order-of-magnitude speedup is highly beneficial for downstream tasks. Many structure design tasks require repeatedly generating large numbers of samples to assess feasibility or increase diversity, and the reduction in inference time linearly increases the number of samples that can be generated. The up to 24× acceleration on short sequences makes the previously time-consuming sampling process substantially lighter, enabling researchers to explore larger structural spaces under the same computational budget, thereby improving design quality and success rates.
>
> | #Tokens     | AlphaFold3 Avg Time (s) | DCFold Avg Time (s) |
> |-------------|--------------------------|-----------------------|
> | <255        | 92.63                    | 3.76                  |
> | 256–383     | 103.31                   | 5.77                  |
> | 384–511     | 112.35                   | 7.17                  |
> | 512–639     | 126.41                   | 10.87                 |
> | 640–767     | 142.78                   | 14.65                 |
> | 768–895     | 169.20                   | 20.02                 |
> | >896        | 212.12                   | 27.40                 |
>
> -----
> # Q2 Training efficiency of DCFold
> We apologize for the oversight in describing the training details in our work. We used 64 NVIDIA H800 GPUs, i.e., batch size = 64, with Stage 1 (Diffusion Consistency) trained for 40 hours over 9k steps, and Stage 2 (Pairformer Consistency) trained for 7 hours over 1.5k steps. According to [2], the original AlphaFold3 training requires a batch size of 256, with Initial training and Finetune 1–2 totaling approximately 150k steps. It is evident that the additional training overhead of DCFold on top of AlphaFold3 is much smaller than training AlphaFold3 from scratch, with fewer training steps and faster training efficiency as demonstrated in Section 4.3. We will include these details in the revised version to facilitate better reproducibility of our work.

---

> ### Author Response · Authors · 2025-11-19
> **Response to Reviewer XzaU (2/n)**
>
> # Q3 Include the AlphaFold3 baseline
> We sincerely thank the reviewer for the valuable suggestions. In the revised version, we have added the results of the original AlphaFold3 implementation for a more comprehensive comparison. We also emphasize that Dual Consistency is model-agnostic: it is not tied to Protenix and can be applied to any AlphaFold-like architecture, to further enhance its generation capabilities.
>
> Regarding the choice of Protenix + Posebusters V2 as our primary evaluation setting, we note that, according to both the Protenix technical report and our reproduction experiments, Protenix generally achieves stronger performance on Posebusters V2, especially on the Success Rate metric ($\text{RMSD} < 2\text{A}$). Prior work [3] has also observed that Protenix can substantially outperform AlphaFold3 on certain complex prediction accuracy metrics, which supports our use of Protenix as a representative base model. Nevertheless, as shown in the updated table, AlphaFold3 achieves comparable overall performance and even surpasses Protenix on the $\text{RMSD} < 5\text{A}$ metric.
>
> | Method | <1 (Best %) | <2 (Best %) | <3 (Best %) | <5 (Best %) | <1 (Worst %) | <2 (Worst %) | <3 (Worst %) | <5 (Worst %) |
> |-------|-------|------|------|-----|------|------|-------|------|
> | AlphaFold 3 | **67.14**  | 80.95   | 85.71   | **94.76**   | 46.19| 68.57| 78.10| 88.57|
> | Protenix | **67.14** | **82.86**   | **87.14**   | 93.81| 45.71| 70.00| 79.05| 87.62 |
> | Protenix ODE | 51.43  | 74.77  | 83.81 |  92.38| 37.62| 66.19 | 75.71 | 87.62|
> | **DCFold (Ours)** | 58.10 | 78.57  | 86.67   | 94.29  | **46.67**  | **71.43**  | **80.00** | **90.48** |
>
> -----
> # Q4 Discussion on Protenix-mini
> We sincerely appreciate your thoughtful suggestions on our experiments and the deeper insights you have contributed to our work.
>
> For a thorough and comprehensive study of accelerating AlphaFold3 (AF3), we have included the concurrent work Protenix-mini (released in July) in our comparison, and we provide a clearer discussion of its methodological differences and empirical results relative to DCFold.
>
> 1. **Methodologically:** Protenix-mini reduces the number of blocks across AF3 modules, compressing the original 368M-parameter model into a 135M-parameter model, and retrains it accordingly. This parameter-reduction strategy trades capacity for efficiency. In contrast, DCFold focuses on reducing the number of generative steps (i.e., *fewer recycling / diffusion steps*) while preserving the full parameter capacity of the original AF3 architecture. This corresponds to “fewer steps” rather than “smaller models”. Since approaches such as consistency models and flow matching (already adopted in the AF model family) naturally favor step reduction, this strategy enables substantial efficiency gains without compromising representational power.
>
> 2. **Performance:** Significantly reduce the number of parameters inevitably leads to non-trivial performance degradation that is hard to fully eliminate through training. In contrast, DCFold demonstrates that by removing a large number of steps while keeping model capacity intact, it can still surpass AF3 (Table 3). In terms of efficiency, DCFold achieves superior performance with significantly fewer NFE, representing a more favorable balance between speed and accuracy.
>
> To provide the community with a comprehensive and contemporaneous comparison, we have added the Protenix-mini baseline in Figure 3. Protenix-mini was submitted to arXiv on 2025-07-16 and officially announced in its repository on 2025-07-17, making it concurrent with our work. We will include this baseline in the revision to more clearly articulate the novelty and advantages of DCFold.
>
> | Method        | Diffusion NFE | Complex LDDT | Prot-Prot LDDT  | Lig-Prot LDDT  |
> |--------|----------|--------|--------|-----|
> | Protenix-mini | 2| 0.802        | 0.490           | 0.622          |
> | AF3 ODE       | 1 | 0.799        | 0.455           | 0.623          |
> | AF3 TGM       | 1| 0.815        | 0.489           | 0.637          |
> | DCFold (Ours) | 1 | 0.819        | 0.507           | 0.646          |
> | AF3           | 200           | 0.821        | 0.501           | 0.650          |
>
> -----
> # Q6 Definition of in silico success rate
> Thank you for raising this question. We have defined the Success Rate of this experiment in Section 4.2 under the Metrics subsection and in Appendix B.1. Model-based Constraints are derived from the confidence scores, while Physics-based Constraints are obtained from physics-based metrics computed by Rosetta. These metrics ensure that the generated binder structures are geometrically feasible, physically stable, and indeed form the intended interaction interface with the target. Therefore, the Success Rate measures the proportion of binders that the model successfully designs while satisfying structural plausibility and functional requirements, serving as a stringent evaluation metric at the in silico stage.

---

> ### Author Response · Authors · 2025-11-19
> **Response to Reviewer XzaU (3/3)**
>
> # Q7 Include other binder design baselines
> Thank you for your suggestion regarding our experiments, which encouraged us to evaluate our work more comprehensively. We have added the RFDiffusion[4] baseline in Table 4. We used the same binder length range as other baselines, sampled 200 binders per target, and applied the same ProteinMPNN design sequence $\rightarrow$ AF2 monomer re-predict pipeline as well as the same success rate computation.
>
> Based on our survey, FoldFlow-2[5] has not publicly released its conditional generation code and has not been trained on any complex datasets. We therefore believe it is not well-suited for the binder design task and did not include it in our baseline comparisons.
>
> When using RFDiffusion for binder design, extensive sampling, multiple designs, and various filtering steps are often required, as a single sampling cannot simultaneously satisfy multiple physical and biological constraints. Consequently, its performance in our baseline is lower than that of BindCraft and DCFold. Additionally, hallucination-based methods avoid the need to train a new model and can leverage the substantial prior knowledge embedded in the folding model, making BindCraft and DCFold more accessible than RFDiffusion.
>
> Physics-based Constraints
> | Target        | ALK(7NWZ) | H3(3ZTJ) | IL2Ra(1Z92) | LTK(7NX0) | TrkA(2IFG) | VirB8(4O3V) |
> |---------------|-----------|----------|-------------|-----------|------------|-------------|
> |**RFDiffusion**| .00       | .02      | .00         | .01       | .00        | .00         |
> | **BindCraft** | .14       | .16      | .38         | .43       | .29        | .15         |
> | **DCFold**    | .12       | .23      | .37         | .47       | .31        | .21         |
>
> Model-based Constraints
> | Target        | ALK(7NWZ) | H3(3ZTJ) | IL2Ra(1Z92) | LTK(7NX0) | TrkA(2IFG) | VirB8(4O3V) |
> |---------------|-----------|----------|-------------|-----------|------------|-------------|
> |**RFDiffusion**| .03       | .07      | .02         | .04       | .01        | .02         |
> | **BindCraft** | .48       | .52      | .84         | .70       | .88        | .72         |
> | **DCFold**    | .54       | .71      | .79         | .93       | .84        | .85         |
>
> [1] Geng, Zhengyang, et al. "Consistency models made easy." arXiv preprint arXiv:2406.14548 (2024).
>
> [2] Abramson, Josh, et al. "Accurate structure prediction of biomolecular interactions with AlphaFold 3." Nature 630.8016 (2024): 493-500.
>
> [3] Zhou, Fan, et al. "Benchmarking AlphaFold3-like Methods for Protein-Peptide Complex Prediction." bioRxiv (2025): 2025-03.
>
> [4] Watson, Joseph L., et al. "Broadly applicable and accurate protein design by integrating structure prediction networks and diffusion generative models." BioRxiv (2022): 2022-12.
>
> [5] Huguet, Guillaume, et al. "Sequence-augmented SE (3)-flow matching for conditional protein generation." Advances in neural information processing systems 37 (2024): 33007-33036.

---

### Official Review · Reviewer_9xiz · 2025-10-31

**Soundness:** 3
**Presentation:** 3
**Contribution:** 3
**Rating:** 6
**Confidence:** 3

**Summary:**

This paper introduces DCFold, a single-step protein structure generation model designed to match AlphaFold3-level accuracy while significantly improving inference efficiency. The motivation stems from AlphaFold3’s diffusion-based iterative design, which, despite its accuracy, incurs high computational cost and long inference times, limiting its use in practical protein design and virtual screening workflows.

DCFold proposes a Dual Consistency (DC) training framework that incorporates a novel Temporal Geodesic Matching (TGM) scheduler to maintain structural fidelity without iterative refinement. As a result, DCFold achieves up to 15× faster inference compared to AlphaFold3 while preserving predictive accuracy. The model’s effectiveness is validated on both structure prediction and binder design benchmarks, demonstrating strong performance with substantially reduced computation.

**Strengths:**

- Proposes a single-step protein generation framework that achieves AlphaFold3-level accuracy with a 15× speedup, addressing a key efficiency bottleneck in diffusion-based methods.
- Introduces a novel Dual Consistency training scheme with Temporal Geodesic Matching, providing a principled way to maintain structural fidelity without iterative refinement.

**Weaknesses:**

- Lacks comparison with other training-based distillation methods such as Protenix-mini, which also reduce inference cost by compressing model size. It remains unclear how “fewer steps” (as in DCFold) compares to “smaller models” in terms of trade-offs between efficiency and performance.
- The **PoseBusters V2 benchmark** results report “best” and “worst” performance without clarifying how these were derived; additional explanation would improve interpretability.
- The paper provides limited details on the **binder design task** setup. It is unclear how much faster DCFold is compared to *BindCraft*, and how many samples were used to compute the *in silico* success rates.

**Questions:**

Q1 How does distilling to a single-step model affect aspects such as the confidence model calibration and sampling diversity?
Q2 I found the purpose of Proposition 1 somewhat unclear—could the authors clarify its specific role in the later theoretical derivations or proofs?

---

> ### Author Response · Authors · 2025-11-19
> **Response to Reviewer 9xiz (1/n)**
>
> # W1 Discussion on Protenix-mini
> We sincerely appreciate your thoughtful suggestions on our experiments and the deeper insights you have contributed to our work.
>
> For a thorough and comprehensive study of accelerating AlphaFold3 (AF3), we have included the concurrent work Protenix-mini (released in July) in our comparison, and we provide a clearer discussion of its methodological differences and empirical results relative to DCFold.
>
> 1. **Methodologically:** Protenix-mini reduces the number of blocks across AF3 modules, compressing the original 368M-parameter model into a 135M-parameter model, and retrains it accordingly. This parameter-reduction strategy trades capacity for efficiency. In contrast, DCFold focuses on reducing the number of generative steps (i.e., *fewer recycling / diffusion steps*) while preserving the full parameter capacity of the original AF3 architecture. This corresponds to “fewer steps” rather than “smaller models”. Furthermore, diffusion models already benefit from several acceleration techniques, such as consistency models and flow matching (which have been extensively used in improvements over AlphaFold3). These techniques enable diffusion models to reduce the number of sampling steps without compromising the model’s expressiveness.
>
> 2. **Performance:** Protenix-mini reduces the model’s parameter count to obtain a lightweight architecture, thereby lowering inference latency. In contrast, DCFold demonstrates that by removing a large number of steps while keeping model capacity intact, it can still surpass AF3 (Table 3). In terms of efficiency, DCFold achieves superior performance with significantly fewer NFE, representing a more favorable balance between speed and accuracy.
>
> To provide the community with a comprehensive and contemporaneous comparison, we have added the Protenix-mini baseline in Figure 3. Protenix-mini was submitted to arXiv on 2025-07-16 and officially announced in its repository on 2025-07-17, making it concurrent with our work. We will include this baseline in the revision to more clearly articulate the novelty and advantages of DCFold.
>
> | Method        | Diffusion NFE | Complex LDDT | Prot-Prot LDDT  | Lig-Prot LDDT  |
> |---------------|---------------|--------------|-----------------|----------------|
> | Protenix-mini | 2             | 0.802        | 0.490           | 0.622          |
> | AF3 ODE       | 1             | 0.799        | 0.455           | 0.623          |
> | AF3 TGM       | 1             | 0.815        | 0.489           | 0.637          |
> | DCFold (Ours) | 1             | 0.819        | 0.507           | 0.646          |
> | AF3           | 200           | 0.821        | 0.501           | 0.650          |
>
> -----
> # W2 The method for computing the best and worst samples
> We thank the reviewer for bringing this up and helping us improve the clarity of our manuscript. As the reviewer correctly understood, the values in Table 2 are computed with respect to the ground-truth structures. However, this does **not** constitute any form of data leakage, because we do not use the ground-truth structures for any filtering or post-processing; instead, it is simply used to measure the range of the minimum and maximum $\text{RMSD}$ achieved by the sampled structures.
>
> To clarify the evaluation protocol: in Section 4.1, DCFold generates 5 samples for each test entry under the same random seed, computes the $\text{RMSD}$ to the ground-truth structure for each sample, and ranks these samples accordingly. The lowest $\text{RMSD}$ is reported as the *best*, and the highest as the *worst*. Because Table 2 aims to compare the **distributional differences** between AlphaFold3’s and DCFold’s predictions—rather than to assess the **correlation between confidence and structural quality**—using ground truth here does not leak information into the model. Our intention is to show that DCFold exhibits a higher upper bound on sample quality and a higher lower bound compared to AlphaFold3, demonstrating DCFold’s ability to avoid low-quality predictions.
>
> Regarding confidence prediction, DCFold inherits AlphaFold3’s performance because the confidence module is frozen during training. As shown in our response to Question 1, a direct comparison between AlphaFold3 and DCFold confidence scores confirms that DCFold maintains AlphaFold3’s capability in confidence estimation.

---

> ### Author Response · Authors · 2025-11-19
> **Response to Reviewer 9xiz (2/n)**
>
> # W3 Details of the binder design experiment
> Thank you for your valuable suggestions regarding our experimental section. We conducted experiments on a single H800 GPU. On the targets used in Table 4, the average GPU time for one full hallucination with BindCraft is 138s, while DCFold requires 105s. Since we follow the same pipeline as BindCraft, the total serial runtime also includes the time for ProteinMPNN and the re-prediction step in addition to the design model’s GPU time. We also provide the total number of designs generated in our experiments in the table below. Overall, DCFold attains slightly better efficiency while producing a comparable number of samples, ensuring a fair comparison.
>
> As mentioned in our responses to Reviewer XzaU – Question 1 and Reviewer KePm – Weakness 1 regarding the relationship between sequence length and runtime, DCFold’s performance gains are even more pronounced on simpler targets (e.g., VirB8).
>
> | Target        | ALK(7NWZ) | H3(3ZTJ) | IL2Ra(1Z92) | LTK(7NX0) | TrkA(2IFG) | VirB8(4O3V) |
> |---------------|------------|-----------|--------------|------------|-------------|--------------|
> | **BindCraft**  | 188        | 269       | 312          | 348        | 243         | 347          |
> | **DCFold**     | 177        | 295       | 375          | 402        | 256         | 439          |
> -----
> # Q1 The impact of dual consistency on confidence and diversity
> Thank you for your insightful questions, which helped us further investigate the performance advantages of DCFold. We have added the Diversity and Confidence metrics for both AlphaFold3 and DCFold on Posebusters V2, and found that there is no significant shift in terms of diversity and confidence. For each sequence, we sample 5 structures and compute the pairwise TM-score among these structures; the dataset-level average serves as the Diversity metric (lower is better). We compute the pLDDT of all sampled structures as the Confidence metric (higher is better).
>
> Our experiments show that, after Dual Consistency training, DCFold exhibits no substantial changes in these metrics compared with AlphaFold3: diversity becomes slightly worse while confidence increases slightly. This indicates that the process of aligning representations/structures in Dual Consistency tightens the structure distribution to some extent while preserving high-quality predictions. This observation is consistent with our statement in Section 4.1 that “such a redistribution reduces extreme errors ...”.
>
> We further conducted experiments by sampling more structures under the same random seed and sampling under different seeds. We observed that neither AlphaFold3 nor DCFold exhibits noticeable improvements in diversity. In our response to Reviewer KePm’s Weakness 3, we discussed the strong conditionality of the AlphaFold family of models, which we believe is closely connected to the difficulty of improving their sampling diversity. It is worth noting, however, that due to the intrinsic conformational heterogeneity of proteins, many prior works have focused on enhancing the sampling diversity of AlphaFold-like models, including but not limited to sampling MSAs, clustering and masking MSA columns, and tuning dropout rates [1–3]. Our acceleration method is orthogonal to most existing diversity-enhancing strategies. These prior techniques can also be applied to DCFold, and we expect them to yield similar improvements as those previously observed on AlphaFold3. We again thank you for raising these questions, which enabled a deeper discussion of this topic.
>
> | Method   | Diversity (↓)       | Confidence (↑)      |
> |----------|---------------------|---------------------|
> | AF3 (5 samples)      | 0.9646 ± 0.0410     | 93.97 ± 2.92        |
> | AF3 (15 samples)     | 0.9642 ± 0.0415        | 93.95 ± 2.93           |
> | AF3 (5 seeds $\times$ 1 sample)        | 0.9697 ± 0.0421        | 93.90 ± 3.01           |
> | DCFold (5 samples)  | 0.9701 ± 0.0565     | 94.14 ± 2.97        |
> | DCFold (15 samples)  | 0.9708 ± 0.0567        | 94.13 ± 2.96           |
> | DCFold (5 seeds $\times$ 1 sample)     | 0.9712 ± 0.0570        | 94.15 ± 2.97           |

---

> ### Author Response · Authors · 2025-11-19
> **Response to Reviewer 9xiz (3/3)**
>
> # Q2 The significance of Proposition 1
> Thank you for raising this point. As you correctly noted, Proposition 1 is not a strictly necessary step for deriving the TGM scheduler. However, as stated in Section 3.3, the starting point of TGM is to address the challenge of Consistency training on variable-length data. The first step in addressing this challenge is to select a distance measure between two distributions that is sensitive to sequence length. For this reason, we adopt the KL divergence, and we hold the belief that “a stable and efficient distillation process must be grounded in the Kullback–Leibler (KL) divergence.”
>
> In Proposition 1, we prove that the geodesic distance between timesteps (t) and (r) reflects the KL divergence between the corresponding distributions on ($\mathcal{M}_t$). Moreover, Proposition 1 shows that the geodesic distance approximates the KL divergence with high accuracy, incurring only a cubic-order error. This confirms that the geodesic distance ($d_g(t, r)$) is indeed the correct metric for measuring separation between timesteps, rather than an ad hoc choice.
>
> [1] Wayment-Steele, Hannah K., et al. "Predicting multiple conformations via sequence clustering and AlphaFold2." Nature 625.7996 (2024): 832-839.
>
> [2] Wallner, Björn. "AFsample: improving multimer prediction with AlphaFold using massive sampling." Bioinformatics 39.9 (2023): btad573.
>
> [3] Kalakoti, Yogesh, and Björn Wallner. "AFsample2 predicts multiple conformations and ensembles with AlphaFold2." Communications Biology 8.1 (2025): 373.

---

### Author Response · Authors · 2025-12-01
**Summary of Responses and Revisions During the Discussion Period**

We sincerely appreciate the constructive feedback provided by the reviewers during the discussion phase. The reviewers largely recognized:

1. **The satisfactory performance of our proposed DCFold**, including its comparable accuracy relative to AlphaFold3 and its promising inference speedup *(All Reviewers)*, which together help unlock the applicability of the AlphaFold3 family of models in downstream tasks *(Reviewer M62B)*.
2. **The novelty of our proposed Dual Consistency training framework for folding models** *(All Reviewers)*.

In our responses to the reviewers, we have already:

1. **Conducted more experiments to provide additional comparisons** on both the structure prediction and binder design tasks *(Reviewers 9xiz, XzaU)*.
2. **Provided fine-grained illustration of our methodology in greater detail** with regard to its effectiveness and efficiency *(Reviewers XzaU, KePm)*.
3. **Incorporated more clarifications to improve the clarity and accessibility of our work**, including additional details on the experimental setup *(Reviewers 9xiz, XzaU, KePm)* and refined descriptions throughout the manuscript *(Reviewer M62B)*.

We are pleased to know that we have **fully addressed all concerns raised by Reviewer M62B**, who subsequently increased the rating. However, due to recent policy changes, it is unfortunate that we were unable to engage in more in-depth discussions with the other reviewers. We believe that we have strengthened the paper through the revisions and have carefully addressed all remaining issues. We sincerely thank the Area Chairs and all reviewers for their efforts and understanding.

---

### Meta-Review · Area_Chair_pwtA · 2026-01-06

**Summary:**

This paper presents DCFold, a distillation method which uses consistency training to reduce the number of pairformer recycling steps and diffusion model steps for Alphafold3-like models. This greatly speeds up inference and is especially interesting with respect to BindCraft-type design methods. Overall I think this is potentially an extremely impactful work as there are many applications of accelerated inference for AF3 style models. Therefore I recommend this paper for acceptance as an Oral presentation.

* Lack of comparison to Protenix-mini (especially smaller models vs. fewer steps)
* Lack of granularity in inference time analysis
* Unclear benchmarking in PoseBusters "best" and "worst" definitions
* Additional clarity and writing concerns

**Reviewer Concerns:**

All concerns were addressed adequately.
* Additional comparisons to Protenix-mini (smaller model) and original AF3
* Added granularity in inference time analysis over token counts
* Added clarifications throughout

**Reviewer Scores:**

9xiz initially rated 6, citing concerns with lack of comparisons, clarification of the PoseBusters benchmark, and limited context for the binder design task. I found the response quite compelling. I would have bumped my score to an 8

XzaU initially rated 8 and I think would have kept their very positive initial impression

KePm initially rated 4 citing lack of granularity in inference time, clarification on PoseBuster's benchmark, and some other comments around clarification. I believe the main weaknesses were addressed sufficiently. I would have bumped the score to 6 or 8 as a reviewer in this position.

M62B commented that they raised their score to 6 after the rebuttal. I agree with this assessment.

---

### Decision · Program_Chairs · 2026-01-26

Accept (Oral)